# Age-Related Increases in PDE11A4 Protein Expression Trigger Liquid–Liquid Phase Separation (LLPS) of the Enzyme That Can Be Reversed by PDE11A4 Small Molecule Inhibitors

**DOI:** 10.3390/cells14120897

**Published:** 2025-06-13

**Authors:** Elvis Amurrio, Janvi H. Patel, Marie Danaher, Madison Goodwin, Porschderek Kargbo, Eliska Klimentova, Sonia Lin, Michy P. Kelly

**Affiliations:** 1Department of Neurobiology, University of Maryland School of Medicine, Baltimore, MD 21201, USA; 2Center for Research on Aging, University of Maryland School of Medicine, Baltimore, MD 21201, USA

**Keywords:** phosphodiesterase, hippocampus, PDE11A, memory, aging, cognitive impairment, biomolecular condensation, tadalafil, inclusion body

## Abstract

PDE11A is a little-studied phosphodiesterase sub-family that breaks down cAMP/cGMP, with the PDE11A4 isoform enriched in the memory-related hippocampal formation. Age-related increases in PDE11A expression occur in human and rodent hippocampus and cause age-related cognitive decline of social memories. Interestingly, age-related increases in PDE11A4 protein ectopically accumulate in spherical clusters that group together in the brain to form linear filamentous patterns termed “PDE11A4 ghost axons”. The biophysical/physiochemical mechanisms underlying this age-related clustering are not known. Here, we determine if age-related clustering of PDE11A4 reflects liquid–liquid phase separation (LLPS; biomolecular condensation), and if PDE11A inhibitors can reverse this LLPS. We show human and mouse PDE11A4 exhibit several LLPS-promoting sequence features, including intrinsically disordered regions, non-covalent pi–pi interactions, and prion-like domains that were particularly enriched in the N-terminal regulatory region. Further, multiple bioinformatic tools predict PDE11A4 undergoes LLPS. Consistent with these predictions, aging-like PDE11A4 clusters in HT22 hippocampal neuronal cells were membraneless spherical droplets that progressively fuse over time in a concentration-dependent manner. Deletion of the N-terminal intrinsically disordered region prevented PDE11A4 LLPS despite equal protein expression between WT and mutant constructs. 1,6-hexanediol, along with tadalafil and BC11-38 that inhibit PDE11A4, reversed PDE11A4 LLPS in HT22 hippocampal neuronal cells. Interestingly, PDE11A4 inhibitors reverse PDE11A4 LLPS independently of increasing cAMP/cGMP levels via catalytic inhibition. Importantly, orally dosed tadalafil reduced PDE11A4 ghost axons in old mouse ventral hippocampus by 50%. Thus, PDE11A4 exhibits the four defining criteria of LLPS, and PDE11A inhibitors reverse this age-related phenotype both in vitro and in vivo.

## 1. Introduction

After the age of 60, nearly all individuals experience some form of cognitive decline—particularly memory deficits—and no drugs prevent or reverse this loss [1,2]. Advanced age is the strongest risk factor for dementia (e.g., [3]), but even in the absence of dementia, age-related cognitive impairment increases healthcare costs and risk for disability [4]. Literature suggests intracellular cAMP and cGMP signaling are decreased in the aged and demented human and rodent hippocampus [5,6,7]. Age-related increases in phosphodiesterase 11A4 (PDE11A4), an enzyme that breaks down cAMP/cGMP, are thought to contribute to these hippocampal signaling deficits and appear to be a fundamental mechanism underlying select long-term memory deficits associated with age-related cognitive decline [8,9,10]. That is, preventing age-related increases in PDE11A4 expression from occurring in mice prevents age-related cognitive decline of social memories [9], and reversing age-related increases in PDE11A4 protein in old mice reverses their memory deficits [8].

PDE catalytic activity does not simply control the total cellular content of cyclic nucleotides, it generates individual subcellular signaling compartments [11,12]. Such subcellular compartmentalization of cyclic nucleotides allows a single cell to respond discretely to diverse intra- and extracellular signals [13]. Thus, the aberrant localization of a PDE has the potential to be far more damaging than a simple loss of catalytic activity [13]. Such mislocalization would not only remove a PDE from its normal pool of cyclic nucleotides (i.e., loss-of-function), it would potentially displace another PDE and ectopically hydrolyze a foreign pool of cyclic nucleotides (i.e., gain-of-function since different PDEs have differing catalytic activities). It is then quite interesting that age-related increases in PDE11A4 protein expression do not occur uniformly throughout the hippocampus, but rather are ectopically enriched in the ventral hippocampus (also known as anterior hippocampus in primates) membrane fractions and “ghost axons”—that is, axonal terminals that are so densely packed with PDE11A4 protein that other axonal markers are occluded [9].

Little is understood of the biophysical/physiochemical mechanisms regulating PDE11A4 age-related clustering, but liquid–liquid phase separation (LLPS) presents itself as a likely candidate. LLPS (also known as biomolecular condensation or inclusion body formation) represents a reversible de-mixing event implicated in neurodegenerative disorders and aging [14]. Depending on the molecule, LLPS can sequester unneeded protein, buffer proteins (i.e., temporarily store and then release upon demand), or accelerate biochemical reactions by virtue of concentrating enzymes with substrates in membraneless organelles [15]. LLPS is driven by either heterotypic condensation (i.e., interaction between client and a scaffolding protein/RNA) or homotypic condensation (i.e., self-association), the latter of which is particularly well nucleated by intrinsically disordered regions (IDRs) [14,16]. Interestingly, the regulatory N-terminal domain of both human (hPDE11A4 [17]) and mouse PDE11A4 (mPDE11A4 [18]) contains an IDR. Further, phosphorylation of serines 117 and 124 (pS117/pS124) within this IDR promotes age-related clustering of PDE11A4 [9] and phosphorylation of residues within IDRs is known to regulate LLPS [16,19]. These physiochemical properties, in addition to the fact that PDE11A4 clusters (1) are largely spherical, (2) increase as endogenous expression increases, (3) require PDE11A4 homodimerization, and (4) fail to colocalize with typical organelle markers either in vitro or in vivo [8,9,20], strongly suggest that age-related clustering of PDE11A4 is caused by LLPS. If so, it may be important to identify therapeutic approaches capable of reversing this LLPS. Thus, we sought to determine (1) if/how PDE11A4 undergoes LLPS, and (2) if PDE11A4 catalytic inhibitors (PDE11Ais) can reverse age-related clustering of PDE11A4 protein in vitro and in vivo. Here, we report that age-related clustering of PDE11A4 meets the criterion for LLPS, and that this LLPS is reversed by PDE11is across multiple scaffolds both in vitro and in vivo.

## 2. Methods

### 2.1. LLPS Prediction

To determine if (1) PDE11A4 is likely to have additional IDRs outside of the N-terminal and (2) if any IDRs exhibit LLPS-promoting features, we used the following recommended [14,21,22] computational screening tools: D2P2, https://d2p2.pro/search (accessed on 29 August 2023); PScore, https://pound.med.utoronto.ca/JFKlab/Software/psp.htm (accessed on 29 August 2023); PLAAC, http://plaac.wi.mit.edu (accessed on 29 August 2023); CIDER, https://pappulab.wustl.edu/CIDER/analysis (accessed on 29 August 2023); PhaSePred, http://predict.phasep.pro (accessed on 31 August 2023); and Phase Separation Predictor; http://www.pkumdl.cn:8000/PSPredictor (accessed on 1 August 2024).

### 2.2. Plasmid Generation

Plasmids were generated as previously described [20]. Briefly, Genscript (Piscataway, NJ, USA) generated constructs expressing either emerald-green fluorescent protein (GFP) alone containing an A206Y mutation to prevent GFP dimerization [23] or the mouse *Pde11a* (NM_001081033) sequence fused at the N-terminal with GFP. Note that mouse *Pde11a* is ~95% homologous and the same length as human *Pde11a4* and so the protein is referred to herein as mPDE11A4 for clarity. These constructs were initially generated on a pUC57 backbone and then subcloned into a pcDNA3.1+ mammalian expression vector (Life Technologies; Waltham, MA, USA).

### 2.3. Compounds

Tadalafil (#6311 Tocris, Minneapolis, MN, USA), BC11-38 (#Hy108618 medchem express, Monmouth Junction, NJ, USA), rolipram (#R6520 Sigma, Rockville, MD, USA), papaverine (#P3510 Sigma, Rockville, MD, USA), 8-Bromoguanosine 3′,5′-cyclic monophosphate sodium salt (#B1381 Sigma, Rockville, MD, USA), Sp-8-pCPT-cGMPS (#C 014-05 Biolog, Hayward, CA, USA), 8-Bromoadenosine 3′-5′-cyclic monophosphate sodium salt (#B7880 Sigma, Rockville, MD, USA), Sp-5,6-DCI-cBIMPS (#D 014-05 Biolog, Hayward, CA, USA), Rp-8-pCPT-cGMPS (#C 013-01 Biolog, Hayward, CA, USA), Rp-8-pCPT-PET-cGMPS (#C 046-01 Biolog, Hayward, CA, USA), Rp-8-Br-cAMPS (B 001-05 Biolog, Hayward, CA, USA), and Rp-cAMPS (#BML-CN135-0001 Enzo, Farmingdale, NY, USA) were obtained from commercial vendors and dissolved in DMSO at a stock concentration of 0.01 M. 1,6-hexanediol was obtained from Sigma (#240117; Rockville, MD, USA) and diluted in DMSO at a stock concentration of 100%. For in vitro studies, all stock PDE inhibitors were diluted in supplemented media and 1,6-hexanediol was dissolved in Hanks’ Balanced Salt Solution (HBSS; #14170-112 GIBCO, Grand Island, NY, USA) at a 5% working solution (i.e., 0.423 M). For in vivo studies, tadalafil powder was mixed with peanut butter (see more below).

### 2.4. Cell Culture and Transfection

As described above, PDE11A4 expression in the brain is enriched in the hippocampus [10,24,25]. Therefore, we use the HT22 hippocampal cell line (sex undefined) for most in vitro investigations, with COS1 cells used solely for live-imaging given their larger size and HEK293T cells used solely in 1,6-hexanediol studies as per the existing literature [26]. Notably, both COS-1 and HEK293T cells exhibit similar mPDE11A4 clustering phenotypes as HT22 cells and GFP-mPDE11A4 expression and subcellular compartmentalization in cell culture is equivalent to that observed with endogenous mPDE11A4 [9,20]. As previously described [9,27], cells were maintained in T-75 flasks in Dulbecco’s Modified Eagle Medium (DMEM) with sodium pyruvate (GIBCO, Gaithersburg, MD, USA or Corning, Manassas, VA, USA), 1% Penicillin/Streptomycin (P/S) (GE Healthcare Life Sciences; Logan, UT, USA), and 10% fetal bovine serum (FBS; Atlanta Biologicals, Flowery Branch, GA, USA), with incubators set to 37 °C/5% CO_2_. Cells were passaged at ~70% confluency using TrypLE Express (GIBCO; Gaithersburg, MD, USA).

On the day before transfection, HT22 cells in DMEM + FBS + P/S were plated in 60 mm dishes, 24-well plates, or coverslips depending on the experiment. HT22 and COS1 cells were plated in 35 mm MatTek slide bottom dishes for live imaging (FSH-NC1843727-PK Fischer Scientific; Ashland, MA, USA). HEK293T cells were plated in collagen-coated 24-well plates (Corning, #354408). On the day of transfection, the media were replaced with Optimem (GIBCO, Gaithersburg, MD, USA), and cells were transfected using 5 microliters of Lipofectamine 2000 (Invitrogen; Carlsbad, CA, USA) and 1.875 ug of plasmid DNA per 5 mL of Optimem as per the manufacturer’s protocol. Approximately 19 h post-transfection, the Optimem/Lipofectamine solution was replaced with DMEM + FBS + P/S (i.e., supplemented media). Cells continued growing for five hours in the supplemented media and then followed 1 of 3 courses. They were either (1) stained with 2 μM BODIPY-568 (Invitrogen, #D3835) in 1X phosphate-buffered saline (PBS 10XPowder Concentrate, Fisher BioReagents, Fair Lawn, NJ, USA) for 15 min and then fixed in 4% paraformaldehyde (Sigma Aldrich, Rockville, MD, USA) and mounted using DAPI Fluoromount-G mounting media (Southern Biotech, Birmingham, AL, USA), (2) directly fixed in 4% paraformaldehyde (Sigma-Aldrich, Rockville, MD, USA) in 1X PBS and then stored in 1X PBS, (3) pharmacologically treated for 1 h or 24 h and then fixed as above and stored in 1X PBS, or (4) treated for 1 h or 24 h, then switched to compound-free DMEM + FBS + P/S for 5 h (i.e., washed out) and then fixed and stored in 1X PBS.

Images were captured via a variety of microscopes. Images of cells labeled with Bodipy-568 were captured using the Leica SP8 DM6 CFS confocal microscope (Deerfield, IL, USA) in the University of Maryland School of Medicine (UMSOM) Department of Neurobiology Imaging Core equipped with the Leica TCS SP8 laser system, Leica STP8000 control panel, Lumencor Sola Light Engine epifluorescent lamp, and Leica CTR6 power source using a 63X/1.40 Oil CS2 ∞/0.17/OFN25/E HC PL APO objective. HT22 cells were live imaged using an Olympus VivaView LCV110 fluorescence incubator microscope in the UMSOM Imaging core that was equipped with a 40× UlanSApo 40×/0.95 infinity symbol/0.11-0.23/FN26.5 objective and a Hamatsu Orca-R2 C10600 camera (Bridgewater, NJ, USA) operated by MetaMorph software v7.7.2 (Molecular Devices, San Jose, CA, USA) set to a 20× digital magnification. Live cell imaging of COS1 cells was conducted in DMEM + FBS + P/S at 37 °C/5% CO_2_ on a Zeiss Axiovert 200M microscope (White Plains, NY, USA) in the University of South Carolina School of Medicine Instrumentation Resource Core Facility (RRID:SCR_024955). The Zeiss was equipped with a VivaTome Optical Train, an MRm AxioCam, a PeCon CTI Controller 3700 and a Fluar 20×/0.75 ∞/0.17 objective. Images were acquired every 30 seconds over the span of 42–120 min. Static images for the quantification of mPDE11A4 compartmentalization into spherical droplets were collected from fixed cells stored in 1X PBS using a Nikon Eclipse TE2000-E Inverted microscope (Melville, NY, USA) via a 10×/0.40 CS2 ∞/0.17/OFN25/A objective equipped with a Photometrics CoolSNAP cf camera and CoolLED pE-300lite LED illuminator. Representative images for each well were captured using MetaVue v6.2r6 software and saved as jpeg files. This Nikon microscope is located in the UMSOM Department of Neurobiology Imaging Core. Over the course of experiments, cells were sporadically tested for yeast, fungal, and bacterial infections (Invitrogen; Cat#:C7028), with negative results always obtained.

### 2.5. PDE Activity Assay

As previously described [27,28], cells were treated for 1 h, after which the media was removed, the cells were harvested in PDE assay buffer (20 mM Tris-HCL and 10 mM MgCl_2_) and homogenized using a tissue sonicator (output control: 7.5; duty cycle: 70; continuous). Samples were then held at 4 °C until processing. Total protein levels were quantified using the DC Protein Assay Kit (Bio-Rad, Hercules, CA, USA) according to the manufacturer’s directions. Then, 3 μg of each sample was processed for cGMP- and cAMP-PDE activity using a radiotracer assay based on [29], with some adjustments [27,28]. Briefly, samples were incubated with 35,000–45,000 disintegrations/minute of [^3^H]-cAMP or [^3^H]-cGMP for 10 min. The reaction was then quenched with 0.1 M HCl and neutralized using 0.1 M Tris. Snake venom was then added to the sample and incubated for 10 min at 37 °C. Samples were then run down DEAE A-25 Sephadex columns previously equilibrated in high salt buffer (20 mM Tris-HCl, 0.1% sodium azide, and 0.5 M NaCl) and low salt buffer (20 mM Tris-HCl and 0.1% sodium azide). After washing the columns four times with 0.5 mL of low salt buffer, the eluate was mixed with 4 mL of scintillation cocktail, and then CPMs were read on a Beckman-Coulter liquid scintillation counter. Two reactions not containing any sample lysate were also taken through the assay to assess background, which was subtracted from the sample CPMs. Data was expressed as CPMs/ug protein.

### 2.6. Quantification of GFP-mPDE11A4 Versus BODIPY-568 Signals

ImageJ/Fiji v1.54p with the Coloc2 plugin was used to conduct colocalization analyses. First, multi-color images were imported and split into separate channels (open multichannel → image → color → split channels). Next, the threshold function was used to isolate droplets from diffuse background labeling (image → adjust → threshold). A region of interest (ROI) was set on the GFP-mPDE11A4 image by using the drawing tool to outline the area within the cell that contained PDE11A4 droplets and then applied to the BODIPY-568 image to ensure alignment (analyze → tools → ROI manager → add[+]). The coloc2 plugin was then run using the GFP-PDE11A4 image as channel 1, the BODIPY-568 image as channel 2, and the mask set to the ROI in channel 1 to calculate Pearson’s correlation coefficient and Manders’ coefficients (M1, M2), with values close to 0 meaning there is no colocalization.

### 2.7. Western Blotting

As previously reported [9], cells were sonicated in ice-cold lysis buffer (0 mM Tris-HCl, pH 7.5; 2 mM MgCl_2_; Thermo Pierce Scientific phosphatase tablet #A32959 and protease inhibitor 3 #P0044) or in PDE assay buffer as described above (samples in Section 3.4.). Protein concentrations were determined by DC Protein Assay kit (Bio-Rad; Hercules, CA, USA) according to manufacturer protocol, and were subsequently equalized across samples. Samples were stored at −80 °C until further processing. For Western blotting, 10 µg of total protein/sample was loaded onto 4–12% NuPAGE Bis-Tris gels (Invitrogen, Waltham, MA, USA) and electrophoresed for one hour at 180 volts. GFP-transfected cell samples were included on all PDE11A4 blots as a negative control. Protein was transferred onto a 0.45 µm nitrocellulose membrane (Amersham, #10600008) for two hours at 100 mA. Membranes were then washed twice in Tris-buffered saline with 0.1% Tween 20 (TBS-T) before staining with Ponceau S to determine total protein loading. Note, Ponceau S was chosen over a housekeeping gene as a loading control based on the best-practice statement of the Journal of Biological Chemistry [30]. Images of the stained membranes were collected to later quantify the optical density of the total protein stain (i.e., spanning ~200 kDa to 10 kDa), and then the membranes were rinsed in TBS-T to remove the stain. Blots to be probed with our custom PDE11A4 antibody (chicken polyclonal; Aves, #1-8113; 1:10,000) were blocked in 5% milk while those to be probed with GFP (rabbit polyclonal; Santa Cruz, #sc8334; 1:2000) were blocked in Superblock Blocking Buffer (ThermoFisher, Cat#37515), each with 0.1% Tween 20. Primary antibodies were incubated overnight at 4 °C. The next day, membranes were washed 4 × 10 min with TBS-T and then incubated for 1 h at room temperature with a secondary antibody (anti-chicken: Jackson Immunoresearch, 103-035-155 at 1:40,000; anti-rabbit: Jackson Immunoresearch, 111-035-144 at 1:10,000). Subsequently, membranes were washed 3 × 15 min in TBS-T. Finally, the membranes were immersed in SuperSignal West Pico Chemiluminescent Substrate as per manufacturer’s directions (ThermoScientific, Waltham, MA, USA), wrapped in clear a plastic sheet protector, and exposed to film. Multiple film exposures were taken to ensure signals were within the linear range, and Ponceau S stain and Western blot optical densities were quantified using ImageJ. To account for variances in film exposure, antibody saturation, chemiluminescence reaction, etc. between blots, Western blot data were normalized to the control condition (e.g., EmGFP-mPDE11A4 + vehicle) on each blot, as previously described (e.g., [10,31,32]).

### 2.8. Quantification of mPDE11A4 Clustering into Punctate Spherical Droplets

As previously described [8,33], all images pertaining to an experiment were quantified by an experimenter blind to treatment using the same computer within the same position in the room, the same lighting conditions, and the same percent zoom. Images were loaded onto a gridded template to facilitate keeping track of count locations within the image, and an experimenter blind to treatment scored each image box by box, with cells along the top and left edges of the entire image not included to follow stereological best practices. Images were quantified in a counterbalanced manner such that 1 picture from each condition was evaluated before moving onto a 2nd image from that condition. The experimenter classified cells as exhibiting either diffuse labeling only or as having punctate droplets with or without diffuse labeling. Data are expressed as the % of the total number of labeled cells that exhibited droplets.

### 2.9. In Vivo Study

In vivo studies were reviewed and approved by the Institutional Animal Care and Use Committee at the University of Maryland, Baltimore with treatment and care conducted as per the National Institutes of Health Guide for the Care and Use of Laboratory Animals (Pub 85-23, revised 1996). As previously described [28], young C57BL6 mice were imported from the National Institute of Aging (NIA) colony and were then bred onsite at the University of Maryland School of Medicine (UMSOM). All mice were group-housed 4-5/cage, with old mice tested at ~18 months old. In keeping with standards of data rigor, the *n* was dictated by an a priori power analysis of our PDE11A4 ghost axon data previously reported in *Aging Cell* [9]. The difference in means between young versus old mice in ventral subiculum was 15.875 with an average standard deviation of 4.61. With a desired power of 0.8 and an alpha of 0.05, our power analysis dictated a sample size of *n* = 3/group. In order to use equal numbers of males and females in each group, we increased the *n* to 4, which increased the predicted power to 0.9. As described by others [34] and us [28], we used peanut butter as a vehicle for oral delivery of tadalafil. Briefly, Jif brand creamy peanut butter was melted to a liquid state in a sterile beaker by stirring it on a warming plate. The liquid peanut butter was either used plain or had a body weight-appropriate amount of tadalafil added to yield a 30 mg/kg dose to mice. The plain peanut butter was then slowly poured into molds using a 3 mL syringe without a needle to avoid bubbles or voids. The rectangular cavities of the mold hold 100 μL of liquid and a sterile razor blade was used to scrape off any excess peanut butter that escaped the cavity. The tadalafil-laden peanut butter was poured similarly into separate tadalafil-designated molds. Molds were then placed onto dry ice for ~20 min to freeze and then were stored in an ultralow freezer until time of use. On the day of testing, pellets were removed from the freezer and immediately placed on dry ice where they remained until the precise time at which a mouse was dosed. Each cage of mice contained 1–2 mice/treatment group to avoid confounds related to cage/location. The order in which treatment groups were harvested from each cage was counterbalanced across cages so no treatment group was always the first or last mouse pulled from the cage. The experimenter was aware of treatment groups during dosing and at the final data analysis; however, the experimenter remained blind to treatment at the time of tissue harvesting, sectioning, imaging and data collection. Mice were food-restricted for 1 night and the next morning (Day 1), all mice were transported to a quiet (~50–52 dB), brightly lit room (~700–800 lux) designated for in vivo studies. Mice were then singly housed in clean plastic cages with no bedding, allowed to habituate for 1 h, and then provided a plain peanut butter pellet for habituation. Typically, mice took ~10–60 min to eat the first pellet. At this point, ad lib food was returned to the home cage. On day 2, mice were again habituated to a plain peanut butter pellet, taking ~7–8 min to consume the pellet. On days 3–30, mice were provided a plain peanut butter pellet (vehicle) or a peanut butter pellet containing an 11 or 110 mg/kg dose of tadalafil. Generally, mice ate their pellets in ~2–3 min. Then, 1 h after consuming their pellet, mice were moved to a second room and killed by cervical dislocation. As previously described [9], tissue was harvested fresh on wet ice, flash-frozen in 2-methylbutane sitting on dry ice, and placed directly on the dry ice to allow for evaporation of the 2-methylbutane. Half-brains were then stored long-term at −80 °C until being processed via immunofluorescence.

### 2.10. Immunofluorescence (IF)

Fresh-harvested, flash-frozen brains were embedded in matrix, cryosectioned in the sagittal plane at 20 µm, and thaw-mounted onto a +/+ slide. As previously described [9,24], frozen tissue was fixed using 4% paraformaldehyde in 1X PBS for 10 min. After fixation, the tissue was washed 3 × 10 min in 1X PBS with bovine serum albumin and triton X-100 (PBT). Overnight incubation was conducted at 4 °C using an antibody cocktail in PBT that included 2 antibodies that detect PDE11A4-pS117/pS124, a post-translational modification found almost exclusively in clustered PDE11A4 [9] (Fabgennix PPD11A-140AP at 1:1000 and Fabgennix PPD11A-150AP at 1:500). The following day, sections were washed 4x in PBT and incubated for 90 min in secondary antibody (Alexafluor 488 AffiniPure Donkey Anti-Rabbit, 1:1000, Jackson Immunoresearch # 711-545-152). The secondary antibody was then washed off with PBT in 3 × 10 min washes. After rinsing the slides 10× in 1X PBS, slides were then treated with TrueBlack Lipofuscin Autofluorescence Quencher (Biotium, catalog # 23007). After, excess TrueBlack was rinsed off with agitation 3× in 1X PBS. Once dry, slides were mounted with DAPI fluoromount (Southern Biotech, #0100-20). Slides were kept covered and refrigerated until imaged using a Leica Model DM6 CFS microscope equipped with the Leica TCS SP8 laser system, Leica STP8000 control panel, Lumencor Sola Light Engine epifluorescent lamp, and Leica CTR6 power source. Representative images were captured with the Leica Application Suite X software v3.7.4 using the HC PL APO 20×/0.74 IMM CORR CS2 objective with max projection images from Z-stacks saved as TIFF files.

### 2.11. Counting Ghost Axons

Endogenous mPDE11A4 ghost axons in the ventral subiculum of hippocampus were quantified by an experimenter blind to treatment. PDE11A4 “ghost axons” are filamentous accumulations of PDE11A4 [9]. Ghost axons were quantified using the consistent application of classification standards with consideration to the continuity of the strand, quality of immunofluorescence brightness, and proximity with other ghost axons. Non-specific or dull structures were omitted, with continuous strands of dots considered a single ghost axon. Data accuracy was maintained by conducting each analysis under consistent room lighting, monitor settings, and digital zoom percentage (75% for all images). Images were quantified using the “Analyze” function in the “ImageJ” application along with the “Multi-Point” tool, which incrementally recorded the number of ghost axons upon each click. The total number of ghost axons was then entered into an .xls spreadsheet for subsequent statistical analysis.

### 2.12. Data Analysis

All between-group analyses were performed using Sigmaplot v11.2. No exclusion criteria were employed and no data points removed. Treatment effects were analyzed by one-way ANOVA (F) or Student’s *t*-test (t) when normality as per the Shapiro–Wilk test and/or equal variance as per Levene’s test passed. When these assumptions were not met, then an ANOVA on ranks (H) was used. Following a significant main effect or interaction, post hoc tests were conducted using Student–Newman–Keuls method and significance was defined as *p* < 0.05. Please note that Sigmaplot provides exact *p*-values for post hoc tests following parametric ANOVAs, but only yes or no to “*p* < 0.05” for post hoc tests following a significant non-parametric ANOVA. Thus, specific *p* values are only reported where available. Data are graphed as mean ± standard error of the mean (SEM). Source data, unadjusted/uncropped images, and a video of images shown in Section 3.2 can be found in the Appendix A.

## 3. Results

### 3.1. PDE11A4 Exhibits Sequence Features Associated with Liquid–Liquid Phase Separation (LLPS)

Only a subset of protein sequences are capable of undergoing LLPS under physiological conditions, and they generally involve multivalent sequences that enable protein interactions with a scaffolding protein (i.e., heterotypic condensation) or within itself (i.e., homotypic condensation; c.f., [14]). These “pro-LLPS” protein sequences include IDRs such as the one in the N-terminal of PDE11A4; however, not all IDRs promote LLPS [14]. IDRs that promote LLPS typically contain additional features including phosphorylation and/or other post-translational modifications, non-covalent pi–pi interactions, prion-like domains, low-complexity regions, and a certain fraction and patterning of charged residues (c.f., [14,21,22]). As such, we utilized several recommended computational screening tools [14,21,22] to determine if (1) PDE11A4 is likely to have additional IDRs outside of the N-terminal and (2) if any IDRs exhibit LLPS-promoting features other than the previously described pS117/pS124 motif that falls at the end of the N-terminal IDR [9]. D2P2, a program that synthesizes analyses from nine different disorder prediction algorithms [35], identifies several IDRs within PDE11A4 that are conserved across human and mouse (Figure 1). That said, only the annotated N-terminal IDR is enriched in ANCHOR domains [36,37]. That is, regions that do not form favorable interactions that sustain stable structures on their own but are capable of sustaining a stable structure when interacting with a globular structure (i.e., a disorder-to-order transition). Analyses for non-covalent pi–pi interactions using PScore [38] and prion-like domains using PLAAC [39] identified a conserved enrichment for both motifs within PDE11A4, but again only within the N-terminal IDR (Figure 1). Analysis for low-complexity regions by SEG [40], as reported by PhaSePred [41], reveals two low-complexity regions in the N-terminal IDR of both human and mPDE11A4 and one in the C-terminal IDR of hPDE11A4 only.

Calculation of the distribution and patterning of charged and hydrophobic residues by CIDER [42] places both human and mouse PDE11A4 in the “R1” category since they are weak polyampholytes (0.1147 and 0.1136, respectively) and weak polyelectrolytes (0.1276 and 0.1297, respectively). Thus, they are predicted to form globules as per the Das Pappu diagram. Interestingly, CIDER analyses suggest the formation of these PDE11A4 globules may be context dependent given the proximity of its score to the R2 category and the fact that their fraction of charged residues (FCR) is 0.2422 and 0.2243, respectively (i.e., just below the 0.25 cutoff to be considered R2). Consistent with the fact that the fraction of positively charged residues (*f*+) is approximately equal to the fraction of negatively charged residues (*f*−), the “net charge per residue” (NCPR) score is close to zero at −0.0129 and −0.016, respectively. This suggests PDE11A4 may behave as disordered globules that are regulated by attractive interactions [16,42]. Finally, CIDER calculates the charge patterning κ scores for human and mPDE11A4 as 0.1727 and 0.1736, respectively, both of which fit comfortably within the range of 0.1–0.3 that is highly predictive of proteins undergoing LLPS [16,42].

**Figure 1 cells-14-00897-f001:**
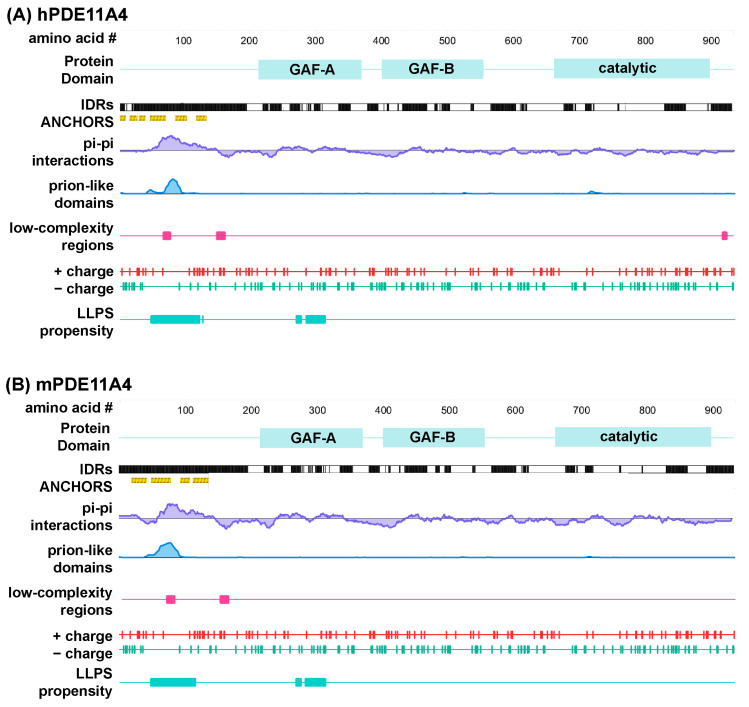
Multiple computational analyses identify LLPS-promoting sequence features within PDE11A4. Graphical outputs from D2P2 [35] (IDRs and ANCHORS) and PhaSePred [41] (all other outputs) for (**A**) human PDE11A4 (hPDE11A4) and (**B**) mouse PDE11A4 (mPDE11A4). PhaSePred reported analyses from other primary computational tools as follows: pi–pi from Pscore [38], prion-like domain from PLAAC [39], low-complexity regions from SEG [40], and LLPS propensity from catGRANULE [43]. GAF—mammalian cGMP-regulated PDEs, Anabaena adenylyl cyclases, and the Escherichia coli transcription factor FhlA; IDRs—intrinsically disordered regions.

In addition to the above algorithms that identify individual LLPS-promoting features, there are additional tools that synthesize the output from several of these tools to calculate an overall likelihood of undergoing LLPS. One such tool is catGRANULE [43], an algorithm that calculates the LLPS propensity of each residue in a given sequence by combining structural disorder, sequence length, nucleic acid binding propensities, and the content of glycine, arginine, and phenylalanine residues [21]. The overall catGRANULE scores for human and mPDE11A4 as reported by PhaSePred [41] are 0.481 and 0.420, respectively. Interestingly, catGRANULE predicts high LLPS propensity for regions of the protein that fall within the N-terminal IDR and the regulatory GAF-A domain (Figure 1). PhaS ePred [41] does not simply report catGRANULE, Pscore, PLAAC, IDR, hydropathy, low-complexity, and charged residues scores all in one place, it also ranks those individual scores amongst all proteins in that species and generates a synthesized ranking for likelihood of undergoing LLPS (0 lowest–1 highest). Strikingly, PhaSePred [41] ranks the Pscore and PLAAC scores for hPDE11A4 at 0.924 and 0.913 out of 1.0, respectively. Further, PhaSePred ranks PDE11A4 as likely to form condensates via both self-assembly (hPDE11A4: score 0.34, rank 0.74; mPDE11A4: score 0.215, rank 0.7) and via interactions with other protein and/or nucleic acid binding partners (hPDE11A4: score 0.46, rank 0.69/1; mPDE11A4: score 0.53, rank 0.73/1). Phase Separation Predictor [44] agrees that both hPDE11A4 and mPDE11A4 will phase separate, each with a score of 0.8861 out of 1.0.

### 3.2. PDE11A4 Exhibits Physical Properties Associated with LLPS

We next determined if PDE11A4 demonstrates physical properties of LLPS. At a minimum, proteins that undergo LLPS should form membraneless spherical droplets that fuse over time in a reversible and concentration-dependent manner [15,19]. To determine if PDE11A4 meets these four criteria, we transfected mouse HT22 hippocampal neuronal cells with mPDE11A4 that was N-terminally fused with emerald-green fluorescent protein (GFP-mPDE11A4) to facilitate microscopic detection. It is highly unlikely that the GFP tag is driving the clustering of mPDE11A4 since (1) immunocytochemistry shows untagged hPDE11A4 similarly forms spherical droplets in vitro [9,33], (2) GFP alone does not form spherical droplets even when expressed at very high levels (Figure 2D), (3) GFP-mPDE11A4 retains its catalytic activity and proper subcellular compartmentalization upon biochemical fractionation [9], and (4) we introduced an A206Y point mutation into our GFP tag to prevent dimerization [23]. Notably, GFP-mPDE11A4 subcellular compartmentalization in cell culture is equivalent to that observed of endogenous hippocampal mPDE11A4 [9,20]. 

We first used GFP-mPDE11A4 to assess if spherical GFP-mPDE11A4 droplets are free of membranes. To do so, we applied BODIPY-568 to label membranes of mouse hippocampal HT22 cells transfected with GFP-mPDE11A4. We quantified the degree of colocalization between PDE11A droplets and the BODIPY signal within 16 cells across four slides using Manders’ overlap coefficients and Pearson’s correlation coefficient. Both methods yielded values close to zero (Manders’ M1 = 0.02 ± 0.01, M2 = 0.03 ± 0.02; PCC r= 0.03 ± 0.02), indicating that there is no measurable colocalization between PDE11A4 and the BODIPY signal (Figure 2A).

Second, we determined if the spherical GFP-mPDE11A4 droplets fuse over time in HT22 neuronal cells. Time-lapsed imaging of four cells over three experiments revealed that GFP-mPDE11A4 spherical droplets emerge over time following a transient transfection and eventually fuse with each other at a rate of 6.5 ± 2.8 fusion events/hour (Figure 2B; see Appendix A for video). We replicated this finding in COS1 cells, which are also known to develop GFP-mPDE11A4 droplets [9]. Emergence and fusion of GFP-mPDE11A4 droplets over time was also observed in seven COS1 cells across five experiments at a rate of 2.43 ± 0.5 fusion events/hour. There was not a significant difference in fusion rates of GFP-mPDE11A4 droplets between HT22 verses COS1 fusion events/hour (T(4,7)-34.0, *p* = 0.0727; Figure 2C). Across cell lines, GFP-mPDE11A4 droplets were noted to separate into smaller droplets on rare occasions. In contrast, larger GFP-mPDE11A4 spherical droplets appeared to become less dynamic over time and could adopt an “irregular shape” that may reflect a liquid–gel phase transition [14]. The fact that GFP-mPDE11A4 droplets fuse over time in both cell types is consistent with the fact that LLPS is driven by the biophysical properties of a protein itself as opposed to cellular processes [45] (Figure 1).

Third, we determined if 1,6-hexanediol, a compound known to reverse LLPS by virtue of interfering with multi-valent interactions (e.g., [46]), would re-mix PDE11A4 droplets. Unfortunately, HT22 cells in our hands did not tolerate this compound, so for this experiment we moved to HEK293T cells plated on collagen-coated plates following the methods of previous reports (e.g., [26]). Again, moving to HEK293T cells for this experiment should not raise concerns since LLPS is driven by the biophysical properties of a protein itself as opposed to cellular processes [45], and 1,6-hexanediol reverses LLPS by disrupting weak hydrophobic interactions that are key drivers of protein LLPS [46]. In a qualitative study, HEK293T cells transfected with GFP-mPDE11A4 were allowed to develop PDE11A4 spherical droplets and were then treated with 5% 1,6-hexanediol (=0.423 M) and live imaged. Within minutes of 1,6-hexanediol being added, GFP-mPDE11A4 spherical droplets dispersed (Figure 2D). Therefore, we conducted a quantitative study in which HEK293T cells transfected with GFP-mPDE11A4 were allowed to develop PDE11A4 spherical droplets and then were treated for 10 min with either vehicle or 5% 1,6-hexanediol. Significantly fewer 1,6-hexanediol-treated cells contained PDE11A4 droplets relative to vehicle-treated HEK293T cells (t(6) = 11.52, *p* = 0 < 0.0001; Figure 2E). Although we could not test 1,6-hexanediol in HT22 cells, it is important to note that we do demonstrate reversibility of GFP-mPDE11A4 spherical droplets in HT22 neuronal cells by another mechanism below.

Fourth, we established if GFP-mPDE11A4 droplets form in a concentration-dependent manner. To do so, we compared GFP-mPDE11A4 droplet formation in hippocampal HT22 cells transfected with a low (0.0375 µg cDNA/mL of media), moderate (0.1125 µg cDNA/mL of media), or high concentration of plasmid (0.375 µg cDNA/mL of media). First, we verified that such titration of cDNA yields low (0.05 ± 0.04 arbitrary units (A.U.)), moderate (0.72 ± 0.2 A.U.), versus high levels (3.82 ± 0.85 A.U.) of GFP-mPDE11A4 protein levels by Western blot (GFP-mPDE11A4 relative optical density normalized by Ponceau S total protein stain; equal variance failed; ANOVA on Ranks: H(2) = 9.85, *p* = 0.0002; post hoc: each vs. the other, *p* < 0.05). Next, we determined if these varying levels of GFP-mPDE11A4 protein would correspond to increasing levels of GFP-mPDE11A4 LLPS. Indeed, the presence of GFP-mPDE11A4 droplets significantly increased as protein levels increased (F(2,9) = 17.96, *p* = 0.0007; post hoc: low vs. moderate *p* = 0.0146, moderate vs. high *p* = 0.0158). A follow-up control experiment confirmed that high expression of GFP alone (i.e., 0.375 µg cDNA/mL of media) did not generate intensely stained droplets like expression of GFP-mPDE11A4 does (image of GFP shown in Figure 2F right-most panel). Whereas 72.2 ± 1.7% of GFP-mPDE11A4-expressing cells were judged to exhibit droplets by a scorer blind to treatment, only 7.2 ± 2.8% of GFP-only-expressing cells were judged to show droplets (*n* = 4/group; t(6) = 19.81, *p* < 0.0001; effect replicated in a second experiment scored by a different scientist). Together, these data show that GFP-mPDE11A4 meets four of the four minimal requirements for LLPS in that it forms membraneless spherical droplets that fuse over time in a reversible and concentration-dependent manner.

### 3.3. PDE11A4’S N-Terminal Intrinsically Disordered Region Is Required for LLPS of the Enzyme

We delved into the mechanism driving PDE11A4 LLPS. As described above, homotypic LLPS is well initiated by intrinsically disordered regions (IDRs), and PDE11A4 contains an N-terminal IDR that is enriched for other pro-LLPS sequence features (Figure 1). As such, we determined if deleting the N-terminal IDR (xN-IDR) would be sufficient to prevent GFP-mPDE11A4 LLPS in HT22 cells. Whereas GFP-mPDE11A4^WT^ undergoes robust LLPS in HT22 cells, GFP-mPDE11A4 ^xN-IDR^ does not (*n* = 5/group; fails normality; Rank Sum: T(5,5) = 40.0, * *p* = 0.0079; representative of 3 experiments). Notably, the lack of GFP-mPDE11A4 ^xN-IDR^ LLPS occurs despite the plasmid producing equivalent GFP-mPDE11A4 protein concentrations relative to GFP-mPDE11A4^WT^ (*n* = 4; WT 1.0 ± 0.14 A.U., xN-IDR 0.93 ± 0.2 A.U.; t(6) = 0.29, *p* = 0.781). Thus, consistent with the prediction algorithms described above, the N-terminal intrinsically disordered region is required for PDE11A4 to undergo LLPS.

### 3.4. PDE11A Inhibitors (PDE11Ais) Inhibit GFP-mPDE11A4 Catalytic Activity in Mouse HT22 Hippocampal Neuronal Cells

Here we report findings with molecules from 2 scaffolds that inhibit PDE11A4. The first scaffold is represented by tadalafil (Figure 3B), a PDE5A inhibitor approved for treatment of erectile dysfunction and benign prostatic hypertrophy that is well known to potently inhibit PDE11A4, albeit at higher concentrations [47,48]. For example, in a pure enzyme assay, tadalafil inhibits PDE5 cGMP-PDE activity with an IC50 of 0.007 µM and PDE11A4 cGMP-PDE activity with an IC50 of 0.05 µM [48]. The second scaffold is represented by BC11-38 (Figure 3B), a PDE11A-selective inhibitor (0.28 µM IC50 in a pure enzyme assay), with >100-fold selectivity for PDE11A4 versus PDE1-10 [49]. We previously verified that tadalafil inhibits GFP-mPDE11A4 cAMP- and cGMP-catalytic activity in mouse HT22 hippocampal cells (reported as “1” in [27]; see Table 1 for means ± SEM). Here, we also confirm the inhibition of GFP-mPDE11A4 in HT22 cells by BC11-38 (Table 1). Both cAMP- and cGMP-PDE activity are very low in GFP-transfected HT22 cells treated with vehicle (i.e., GFP + 0 µM) but are greatly increased in vehicle-treated cells expressing GFP-mPDE11A4 (i.e., WT + 0 µM). This GFP-mPDE11A4-mediated cAMP-PDE activity (F(5,18) = 76.79, *p* < 0.0001; post hoc: WT + 0 µM vs. WT + 100 µM *p* = 0.0004) and cGMP-PDE activity were significantly inhibited by 100 µM BC11-38 (Table 1; F(5,18) = 59.93, *p* < 0.0001; post hoc: WT + 0 µM: vs. WT + 100 µM *p* = 0.0019). Consistent with the short duration of treatment, these reductions in GFP-mPDE11A4 catalytic activity occurred in the absence of any significant changes in GFP-mPDE11A4 protein expression (Table 1).

### 3.5. PDE11Ais Reverse GFP-mPDE11A4 LLPS in Mouse HT22 Hippocampal Cells

We next determined if PDE11Ais were capable of reversing aging-like LLPS of GFP-mPDE11A4 in HT22 cells (Figure 3). We tested both a 1 h and a 24 h treatment to determine whether prolonged exposure would lead to desensitization, a phenomenon often observed with chronic drug treatment due to compensatory mechanisms (Figure 3A). Additionally, we implemented a 5 h washout to assess whether the inhibitors bind reversibly or irreversibly (Figure 3A). A 1 h treatment of either tadalafil (Figure 3D; F(4,15) = 34.18, *p* < 0.0001; post hoc vs. 0 µM: 10 µM and 100 µM, *p* = 0.0002) or BC11-38 (Figure 3C,H; F(4,15) = 5.29, *p* = 0.0073; post hoc 0 µM vs. 100 µM, *p* = 0.0133) reverses GFP-mPDE11A4 LLPS in HT22 cells. Following a 5 h washout of the drug, GFP-mPDE11A4 re-clusters (also known as de-mixes) in the cells treated for 1 h with 10 µM tadalafil (Figure 3E; F(4,15) = 22.75, *p* < 0.0001; post hoc 0 µM vs. 10 µM, *p* = 0.7151) or 100 µM BC11-38 (Figure 3I; equal variance failed; ANOVA on Ranks for effect of compound: H(4) = 7.99, *p* = 0.092), but not in cells treated for 1 h with 100 µM tadalafil (post hoc 0 µM vs. 100 µM, *p* = 0.0002). PDE11A4 inhibitor effects on LLPS were sustained with a 24 h treatment of either tadalafil (Figure 3F; F(4,15) = 28.10, *p* < 0.0001; post hoc: 0 µM vs. 10 µM or 100 µM, *p* = 0.0002 each) or BC11-38 (Figure 3J; F(4,15) = 11.72, *p* = 0.0002; post hoc: 0 µM vs. 100 µM, *p* = 0.0004). Following the 24 h treatment, a 5 h washout allowed GFP-mPDE11A4 to re-cluster in the cells treated with 10 µM tadalafil (Figure 3G; fails equal variance; ANOVA on Ranks: H(4) = 15.16, *p* = 0.0044; post hoc: 0 µM vs. 10 µM, *p* > 0.05) or 100 µM BC11-38 (Figure 3K; effect of compound: F(4,15) = 2.75, *p* = 0.0673). PDE11A4 re-clustering did not occur following a 5 h washout of 100 µM tadalafil (post hoc: 0 µM vs. 100 µM, *p* < 0.05). Importantly, this dispersal of GFP-mPDE11A4 LLPS by PDE11Ais appears to be specific in that neither the PDE4 inhibitor rolipram nor the PDE10 inhibitor papaverine elicited such an effect (Figure 3L,M). Thus, across scaffolds, PDE11Ais reverse GFP-mPDE11A4 LLPS in hippocampal HT22 cells but the effect can be reversed upon compound washout.

**Figure 3 cells-14-00897-f003:**
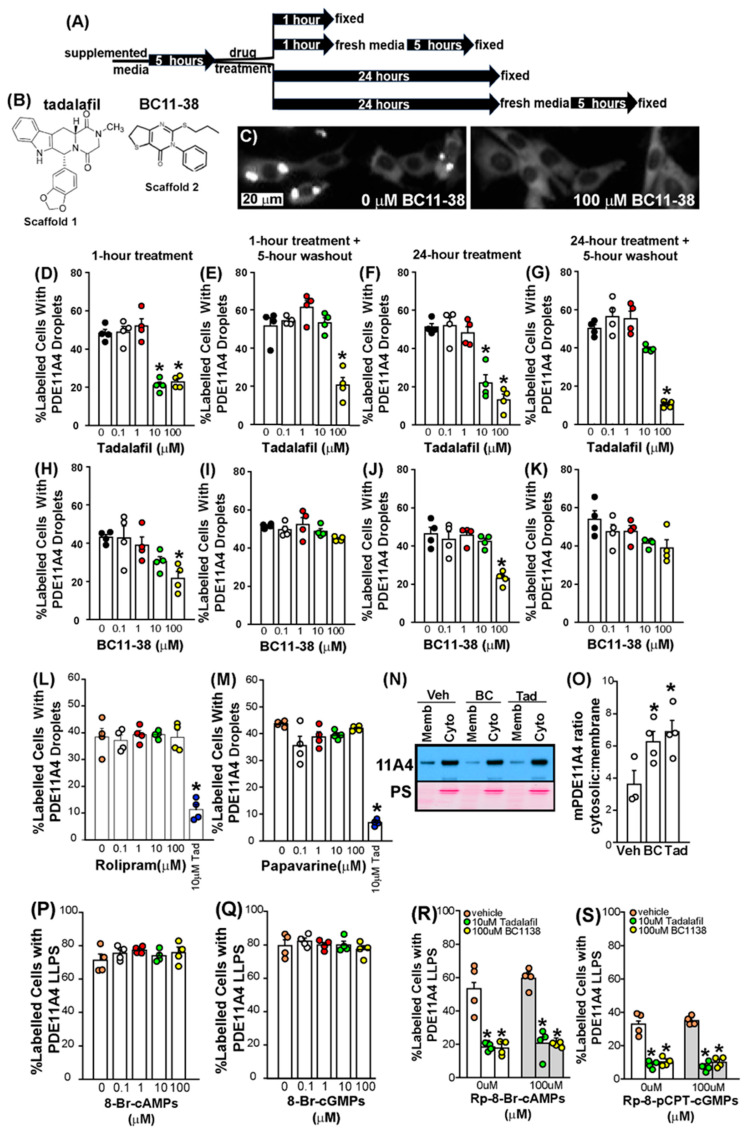
The dispersing effects of PDE11A4 inhibitors do not diminish with a 24 h treatment and the effects can be reversible upon washout. (**A**) A graphical outline of the experimental time course. (**B**) Scaffolds for PDE11is tadalafil and BC11-38. (**C**) Exemplar image of aging-like LLPS of EmGFP-mPDE11A4 following vehicle treatment (left) and dispersal following a 1 h treatment with 100 µM BC11-38. The extent of mPDE11A4 LLPS measured following a 1 h treatment, a 1 h treatment followed by a 5 h washout period, a 24 h treatment or a 24 h treatment followed by a 5 h washout period for (**D**–**G**) tadalafil and (**H**–**K**) BC11-38 *(n* = 4 biological replicates/group; each data set is representative of 2+ independent experiments). The extent of mPDE11A4 LLPS measured following a 1 h treatment of (**L**) the PDE4 inhibitor rolipram or (**M**) the PDE10 inhibitor papaverine (10 µM tadalafil was included as a positive control; data is representative of 2 independent experiments). (**N**) Representative Western blot (top: EmGFP-PDE11A4; bottom: Ponceau Stain (PS)) and (**O**) quantification showing aging-like accumulation of mPDE11A4 in the membrane fraction in vehicle-treated (Veh, *n* = 3 biological replicates) cells that is reversed by a 1 h treatment with BC11-38 (BC, *n* = 4) and tadalafil (Tad, *n* = 4), thereby eliciting an “anti-aging” effect by increasing the mPDE11A4 ratio in the cytosolic fraction (cyto) versus the membrane fraction (memb). The ability of PDE11Ais to reverse PDE11A4 LLPS appears to be independent of their ability to inhibit PDE11A4 catalytic activity because neither (**P**) cAMP nor (**Q**) cGMP analogs mimic the effect of PDE11Ais and neither (**R**) cAMP nor (**S**) cGMP antagonists block the effect of PDE11Ais on LLPS. * vs. 0 µM/vehicle *p* < 0.05–0.0001. Brightness, contrast, and histogram stretch of images are adjusted for graphical clarity.

### 3.6. The Dispersal of PDE11A4 LLPS Is Not Regulated by Cyclic Nucleotide Levels

Given that PDE11A4 breaks down cAMP and cGMP, and tadalafil and BC11-38 increase cAMP and cGMP by virtue of inhibiting this catalytic activity, we parsimoniously hypothesized that PDE11A4 LLPS would be reduced by elevated cAMP and/or cGMP levels. Surprisingly, PDE11A4 LLPS was not altered by 0.1–100 µM of the cGMP analogs 8-Br-cGMPs (Figure 3Q; F(4,15) = 0.60, *p* = 0.6716) or Sp-8-pCPT-cGMPS (see Source Data in Appendix A; *n* = 4/group; F(4,15) = 2.32, *p* = 0.1047). Similarly, PDE11A4 LLPS remained unchanged with 0.1–100 µM of the cAMP analogs 8-Br-cAMPs (Figure 3P; F(4,15) = 0.81, *p* = 0.5366) and Sp-5,6-DCI-cBIMPs (see Source Data in Appendix A; *n* = 4/group; F(4,15) = 1.68, *p* = 0.206). These results suggest that increasing cyclic nucleotide levels does not reduce PDE11A4 LLPS.

To further test this hypothesis, we determined if cyclic nucleotide signaling blockers would prevent the LLPS-dispersing effects of PDE11Ais. Neither the cGMP blockers Rp-8-pCPT-cGMPs (Figure 3S; Two-way ANOVA failed equal variance; Rank Sum Test for effect of Rp-8-pCPT-cGMPs: T(12,12) = 154, *p* = 0.8399) nor Rp-8-pCPT-PET-cGMPs prevented PDE11Ais from reversing PDE11A4 LLPS (see Source Data in Appendix A; Two-way ANOVA failed normality; Rank Sum Test for effect of Rp-8-pCPT-PET-cGMPs: T(12,12) = 132.0, *p* = 0.3122). The cAMP blockers Rp-8-Br-cAMPs (Figure 3R; Two-way ANOVA fails equal variance; Rank Sum Test for effect of Rp-8-Br-cAMPs: T(12,12) = 137.0, *p* = 0.470) and Rp-cAMP similarly failed to prevent PDE11Ais from reversing PDE11A4 LLPS (see Source Data in Appendix A; *n* = 4/group; Two-way ANOVA fails equal variance; Rank Sum Test for effect of Rp-8-Br-cAMPs: T(12,12) = 142.0, *p* = 0.665). Together, these findings show that the ability of PDE11Ais to reverse PDE11A4 LLPS is unrelated to their ability to increase cGMP and cAMP levels and are consistent with the fact that cyclic nucleotide analogs failed to reduce PDE11A4 LLPS above (Figure 3P,Q).

### 3.7. PDE11Ais Shift GFP-mPDE11A4 from the Membrane to the Cytosolic Fraction

Since PDE11Ais reversed aging-like GFP-mPDE11A4 LLPS, we next determined if PDE11Ais would also reverse aging-like changes in the distribution of GFP-mPDE11A4 between the cytosolic versus membrane fractions as a second measure of a change in subcellular localization. Even though LLPS condensates themselves are membraneless, heterotypic LLPS often takes place at the plasma membrane in response to scaffolding proteins being recruited there following receptor activation [50,51]. Consistent with this model, PDE11A4 is not directly inserted into the membrane but rather is indirectly associated with the membrane [9] and PhasePred results above predict PDE11A4 undergoes both homotypic and heterotypic LLPS. Following biochemical fractionation, PDE11A4 protein is expressed at much higher levels in the cytosolic (also known as soluble) fraction relative to the membrane (also known as particulate) fraction in both the young hippocampus and HT22 cells [9]. Interestingly, however, age-related increases in ventral hippocampal mPDE11A4 are specifically localized to the membrane fraction [9], thus decreasing the ratio of cytoplasmic/membrane mPDE11A4. We see here that a 1 h treatment of HT22 cells with 100 μM of either tadalafil (*n* = 4 biological replicates) or BC11-38 (*n* = 4) has an “anti-aging” effect by increasing the GFP-mPDE11A4 cytosolic/membrane ratio relative to vehicle (*n* = 3; F(2,8) = 4.99, *p* = 0.039; post hoc vs. vehicle: tadalafil *p* = 0.0391, BC11-38 *p* = 0.0389; Figure 3N,O). Thus, both the LLPS and biochemical fractionation experiments show PDE11A4 small molecule inhibitors change the subcellular localization of PDE11A4.

### 3.8. Oral Dosing of Tadalafil to Old NIA C57BL6 Mice Reverses Age-Related Clustering of PDE11A4 in Ghost Axons

Finally, we determined if oral administration of a brain-penetrant PDE11A4 inhibitor would reverse age-related clustering of PDE11A4 in ventral hippocampal ghost axons. Tadalafil is orally bioavailable and crosses the blood–brain barrier with a Tmax of ~1 h [52]. For 30 days, mice were orally dosed with either peanut butter alone (i.e., vehicle) or peanut butter pellets laden with either an 11 mg/kg dose or a 110 mg/kg dose of tadalafil in order to yield a brain exposure of 0 μM, 1 μM, or 10 μM, respectively [52]. Consistent with our in vitro results showing 10 μM but not 1 μM tadalafil reversed GFP-mPDE11A4 LLPS (Figure 3), oral administration of 110 mg/kg tadalafil, but not 11 mg/kg, decreased ventral hippocampal PDE11A4 ghost axons by ~50% in old female and male mice (Figure 4; *n* = 2/sex/group; F(2,9) = 4.77, *p* = 0.039; post hoc 110 vs. 0 mg/kg, *p* = 0.045).

## 4. Discussion

Here, we show that aging-like overexpression of PDE11A4 triggers LLPS (also known as de-mixing or biomolecular condensation) of the enzyme, and PDE11A4 small molecule inhibitors reverse this LLPS both in vitro and in vivo. Computational analyses reveal that mPDE11A4 and hPDE11A4 share many “pro-LLPS” sequence features, including—but not limited to—intrinsically disordered regions, non-covalent pi–pi interactions, prion-like domains, and a highly favorable charge patterning κ score (Figure 1). In cells, we show that aging-like GFP-mPDE11A4 spherical clusters do not colocalize with membranes (Figure 2A), emerge spherical in shape with subsequent fusing over time (Figure 2B,C), and increase in frequency with increasing concentrations of the enzyme (Figure 2F). 1,6-hexanediol (Figure 2D,E) and PDE11A4 inhibitors (Figure 3)—but not PDE4 or PDE10 inhibitors—then reverse this aging-like LLPS of GFP-mPDE11A4 (i.e., re-mix the enzyme) in HT22 neuronal cells. Mechanistically, we showed that PDE11A4 LLPS requires its N-terminal intrinsically disordered region, a sequence motif featured in many proteins that undergo LLPS (Figure 2G). Consistent with these in vitro observations, we see in the old mouse hippocampus that age-related clustering of endogenous mPDE11A4 in so-called “ghost axons” often reveals itself upon higher magnification to be a trail of adjacent spherical droplets as opposed to a continuous filament-like structure (Figure 4D). Perhaps more importantly, we also find that this age-related clustering of endogenous mPDE11A4 is reversed by oral administration of tadalafil (Figure 4). Thus, mPDE11A4 exhibits the main features of LLPS [15,19] in that it forms membraneless spherical droplets that progressively fuse over time in a concentration-dependent and reversible manner, and PDE11Ais reverse this aging-like phenotype.

### 4.1. Computational Analyses Identify Numerous LLPS-Promoting Features in Both mPDE11A4 and hPDE11A4 Enriched in the N-Terminal Regulatory Domain

In addition to its previously described N-terminal IDR, we show here that PDE11A4 exhibits many other sequence features that are associated with LLPS [16,19]. Like alpha-synuclein [16], CIDER [42] characterizes PDE11A4 as a weak polyampholyte and weak polyelectrolyte that likely forms globules (i.e., category “R1” on the Das Pappu diagram). That said, the formation of PDE11A4 globules may be context-dependent (i.e., category “R2”) given the fact that its FCR is 0.24223, which is just below the R2 category cutoff of 0.25. Notably, tau is an R2 LLPS protein [16]. Such context dependence would be consistent with the fact that LLPS of PDE11A4 increases with age, perhaps in response to age-related drops in brain pH [53], and catalytic inhibitors can reverse PDE11A4 LLPS (Figure 3 and Figure 4). The suggestion of context-dependent condensation is also consistent with the PhaSePred [41] prediction that PDE11A4 is likely to undergo both homotypic LLPS due to multivalent interactions with itself as well as heterotypic LLPS due to interactions with scaffolding molecules.

Experimental evidence to date aligns with the suggestion that PDE11A4 is capable of both homotypic and heterotypic LLPS under physiological conditions. Native gels of old mouse hippocampus and PDE11A4-transfected cell cultures exhibiting PDE11A4 droplets reveal that PDE11A4 is present both as an isolated homodimer, suggesting homotypic LLPS is possible, and as part of larger macromolecular complexes, suggesting heterotypic LLPS is possible (see Appendix A in [9]). Regarding homotypic condensation specifically, post-translational modifications of IDRs are known regulators of homotypic LLPS [19], and the PDE11A4-pS117/pS124 signal that drives age-related clustering of PDE11A4 falls within its N-terminal IDR [9]. Further, deletion of this N-terminal IDR prevents PDE11A4 LLPS (Figure 2G). It will be of great interest to future experiments to more selectively probe the functional relevance of the pro-LLPS sequence motifs that we identified within the PDE11A4 N-terminal IDR (Figure 1, e.g., prion-like domains and π-π interactions). The fact that age-related LLPS occurs in parallel with more PDE11A4 protein being associated with the ventral hippocampal membrane fraction [9] supports the possibility that PDE11A4 also undergoes heterotypic LLPS. Even though LLPS condensates themselves are membraneless, heterotypic LLPS often takes place at the plasma membrane in response to scaffolding proteins being recruited there following receptor activation [49,50]. Consistent with this model, PDE11A4 is not directly inserted into the membrane but rather is indirectly associated with the membrane [9]. It will be of great interest to future studies to further explore the mechanistic underpinnings of PDE11A4 homotypic vs. heterotypic LLPS.

### 4.2. PDE11A4 Clustering Meets the Minimum Physical Requirements for LLPS

At a minimum, proteins that undergo LLPS should form membraneless spherical droplets that fuse over time in a concentration-dependent and reversible manner [15,19], and we show here PDE11A4 meets all of these criteria (Figure 2 and Figure 3). PDE11A4 spherical droplets being membraneless is highly consistent with previous studies showing PDE11A4 clusters fail to colocalize with any membrane-bound organelle markers tested [9]. No other PDEs have been reported to undergo LLPS to date, but that may not be surprising given that most do not increase in expression with age and many actually decrease in expression across the lifespan [9,10,24,28,31,54,55]. That said, the regulatory R1α subunit of protein kinase activated by cAMP (PKA) undergoes LLPS in response to elevated cAMP levels [26]. The fact that LLPS appears to be responsible for age-related clustering of PDE11A4 is highly interesting because LLPS is gaining interest in the mechanistic study of dementia and neurodegenerative diseases [14,15,53,56,57,58]. In this context, it may be interesting to note that two rare PDE11A variants have been associated with early-onset Alzheimer’s disease [59] and exacerbated age-related increases in PDE11A expression have been associated with dementia in elderly traumatic brain injury patients [9]. Further, many factors that regulate LLPS in general have been reported to change in the aging and AD brain [9,15,16,53,60,61,62,63]. These factors include elevated protein concentrations as well as reduced salt concentration, temperature, and pH [9,53,60,61,62,63]. A possible relationship between aging/AD-related decreases in brain pH and PDE11A4 LLPS are particularly plausible given the established interplay between cAMP/cGMP/PDE signaling and pH-regulated processes (e.g., [64,65,66,67,68]).

### 4.3. PDE11Ai Small Molecule Inhibitors Reverse mPDE11A LLPS

Here we tested both 1 h and 24 h treatments on PDE11A4 LLPS to determine whether prolonged exposure to the PDE11A inhibitors would lead to desensitization, a phenomenon often observed with chronic drug treatment due to compensatory mechanisms (e.g., effector adaptation, compensatory signaling, or degradation of the drug). The potency and efficacy of the structurally distinct PDE11A inhibitors tested here remained stable (Figure 3B–K), suggesting that PDE11A4 inhibitors do not desensitize but rather elicit a stable pharmacodynamic effect over time. Further, the PDE4 inhibitor rolipram and the PDE10 inhibitor papaverine had no effect on PDE11A4 LLPS (Figure 3L,M), arguing for specificity of the PDE11Ai effects. We also conducted washout experiments to determine whether the PDE11A inhibitors bind reversibly or irreversibly. Interestingly, the LLPS effects of the most potent concentration of each inhibitor washed out in just 5 h (i.e., 10 µM tadalafil and 100 µM BC11-38), however the next higher concentration of tadalafil did not wash out. This pattern was observed following either a 1 h or 24 h treatment. This may suggest that PDE11A4 harbors both a high-affinity site that reversibly binds inhibitors and a low-affinity site that either irreversibly binds inhibitors via covalent modifications or has a much slower off-rate binding. There is precedent in the PDE4 family for such a phenomenon, with some isoforms including both a high affinity and low affinity binding site for inhibitors [69]. In this regard, it is interesting to note that PDE11A4 binds cGMP both as a substrate at the catalytic site and at the regulatory GAF-A domain, the latter of which also binds inhibitors of PKG (c.f., [70]). We believe this is particularly intriguing given that catGRANULE predicted the GAF-A domain to have LLPS propensity (Figure 1). Together, these studies provide key insights relevant to the pharmacological properties of these compounds.

We found that PDE11A inhibitor IC50s are substantially higher in our mammalian cell-based assays than they are in a pure enzyme assay [27,28], a well-known phenomenon in the drug screening literature (c.f., [71,72,73,74]). IC50s in pure enzyme assays are much lower because they reflect direct interactions between the enzyme, compound, and substrate in absence of any barriers [72]. Further, concentrations of the enzyme and substrate do not reflect cellular levels. Rather, each is optimized for the assay to read out inhibition. In cell-based assays, IC50s are much higher for many reasons. First, the concentration of enzyme and substrate present in the cell is often much lower/higher, respectively, with both requiring higher concentrations of the compound to measure inhibition [72]. Second, there is the challenge of the compound having to enter and remain in the cell due to the presence of cellular membranes and efflux mechanisms [70,73]. This challenge can be further exacerbated if the enzyme localizes to subcellular compartments where compound access is restricted. This may be particularly relevant for PDE11A4 inhibitors given the fact that PDE11A4 undergoes LLPS in cells, which we hypothesize serves to sequester unneeded enzyme. Compounds also non-specifically bind to various macromolecules (e.g., proteins, lipids, etc.) in the media and cells [73]. Third, compounds can be degraded or metabolized in a cellular context, either of which reduces the amount of compound available to bind the enzyme [73,74,75]. Indeed, PDE inhibitors are characterized as prone to both metabolic enzymes and intracellular esterases, thus reducing their intracellular concentrations [74,75]. Fourth, cell-based assays require complex culture media that reduces the solubility of a compound, whereas enzyme assays use aqueous buffers with a controlled pH and ionic strength in addition to organic solvents and/or detergents that enhance solubility [76]. Fifth, cells express enzyme binding partners and post-translational modifiers that can reduce sensitivity of the enzyme to inhibitors (e.g., via change in conformation or subcellular localization). Again, this may be particularly relevant for PDE11A inhibitors as the conformation of the PDE11A4 N-terminal regulatory domain gates access to the catalytic site where substrates/inhibitors bind (c.f., [70]). Further, we have shown that both homodimerization and phosphorylation of various PDE11A4 residues is sufficient to change the subcellular localization of PDE11A4 and PDE11A4 LLPS [8,9,20].

Importantly, the lowest concentration of tadalafil that was effective in our HT22 cell model (i.e., 10 µM) was also effective in the brain following oral dosing. This suggests that results from our cell model can readily be used for in vivo dose selection based on brain exposures achieved following oral dosing. The fact that tadalafil reversed age-related PDE11A4 ghost axons at a 10 µM brain exposure (i.e., 110 mg/kg) but not a 1 µM brain exposure (i.e., 11 mg/kg) suggests that this effect of tadalafil is mediated via it binding to PDE11A4, as opposed to inhibiting PDE5 since much lower plasma exposures of tadalafil are required for treating erectile dysfunction [77]. The fact that both inhibitors reverse PDE11A4 LLPS independent of increases in cAMP or cGMP signaling also argues against tadalafil altering PDE11A4 LLPS as a consequence of inhibiting PDE5 catalytic activity. That said, we are unable to unequivocally confirm the PDE11A4 LLPS dispersing effect of tadalafil is mediated specifically by binding PDE11A4—as opposed to inhibiting PDE5 or eliciting other general physiological effects—because the endpoint is PDE11A4 itself. This leaves the classic negative control approach of testing the compound in *Pde11a* KO mice to establish specificity of the compound for PDE11A4 unusable. We should also note that this study is limited by the fact that we did not test rolipram or papaverine in vivo as we did in vitro to demonstrate specificity. Still, the ability of tadalafil to reverse PDE11A4 ghost axons in vivo does provide proof of principle for pursuing more selective and potent PDE11A4 inhibitors to reverse this age-related phenotype. Indeed, since such high concentrations of tadalafil may produce side effects, we are currently testing effects of orally dosed SMQ-03-20, the first selective and potent PDE11A4 inhibitor to cross the blood–brain barrier [28].

The mechanism by which PDE11Ais re-mix PDE11A4 remains to be determined. It is not by reducing PDE11A4 protein expression, as we found no effects of the PDE11Ais on PDE11A4 protein expression (Table 1). It is also not due to elevations in cAMP/cGMP levels that occur as a consequence of inhibiting PDE11A4 catalytic activity, since cAMP/cGMP analogs fail to mimic the effect of PDE11Ais and cAMP/cGMP antagonists fail to block the ability of PDE11Ais to remix PDE11A4 (Figure 3P–S). It is then interesting to note that increased cAMP promotes LLPS of the PKA R1α subunit [26] and select PDE4 inhibitors trigger PDE4A4 to adopt an LLPS-like phenotype (i.e., it accumulates in membraneless spherical structures) [78,79,80]. PDE11Ais may ultimately decrease PDE11A4-p117/p124 (that is the post-translational modification that promotes PDE11A4 LLPS [9]). That said, preventing phosphorylation of S117/S124 only partially reduces PDE11A4 LLPS, it by no means eliminates it as do higher concentrations of the PDE11Ais (Figure 3). Another intriguing possibility is that application of the PDE11Ais reduces PDE11A4 packaging/repackaging via the trans-Golgi network since we previously showed that stimulating Golgi packaging increased PDE11A4 LLPS [33]. It might also be possible that the binding of the inhibitor initiates a conformational change in the enzyme that disperses PDE11A4 LLPS (e.g., reducing PDE11A4 homodimerization). We are actively investigating these possibilities and more.

## 5. Conclusions

Depending on the molecule, LLPS can sequester unneeded protein, buffer proteins (i.e., temporarily store and then release upon demand), or accelerate biochemical reactions by virtue of concentrating enzymes with substrates in membraneless organelles [15]. It will be of paramount importance to future studies to parse this out as it relates to PDE11A4 LLPS specifically. For example, expressing cAMP/cGMP sensors in our model systems, particularly sensors targeted to cytosolic versus membrane compartments, may help us dissect the physiological significance of PDE11A4 LLPS. We recently reported that a biologic that disrupts homodimerization of mPDE11A4 reverses GFP-PDE11A4 LLPS in cells as well as age-related clustering of PDE11A4 “ghost axons” in the old mouse hippocampus. This biologic also selectively degrades membrane-associated GFP-mPDE11A4 and rescues age-related social memory deficits [8,20]. Together, these results may suggest that the ability of PDE11Ais to reverse PDE11A4 LLPS and increase the PDE11A4 cytosolic/membrane ratio will add to the therapeutic effects of their enzymatic inhibition.

## Figures and Tables

**Figure 2 cells-14-00897-f002:**
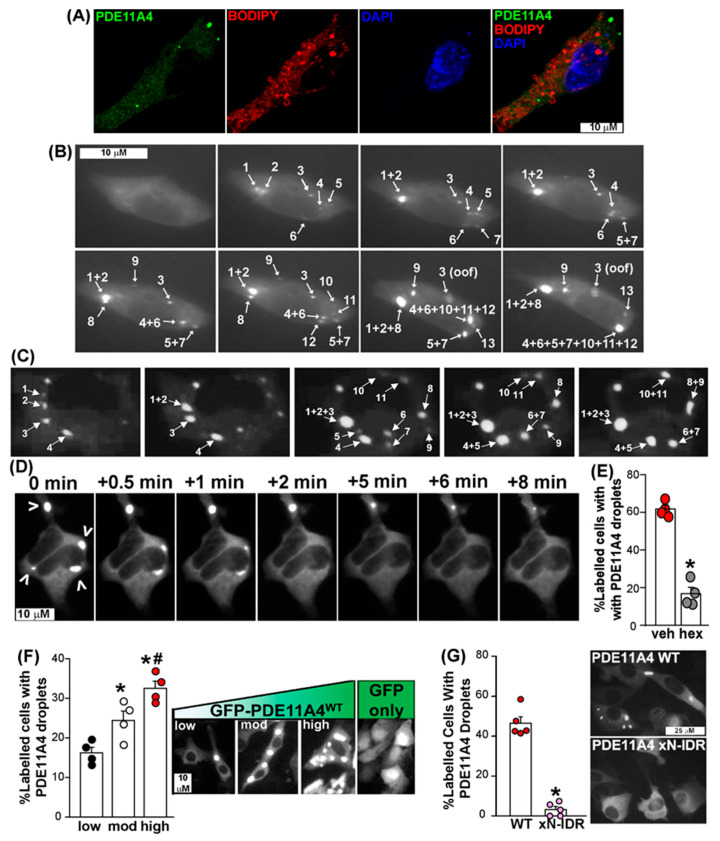
PDE11A4 demonstrates physical features of LLPS in that its punctate accumulation starts as small membraneless spherical droplets that progressively fuse over time in a reversible and concentration-dependent nature. (**A**) Membrane staining using BODIPY-568 in HT22 cells transfected with EmGFP-mPDE11A4 shows no colocalization of the signals (representative image of 16 cells across four coverslips, see Results text for quantification). (**B**) Qualitative time-lapsed imaging of EmGFP-mPDE11A4 in 4 HT22 cells reveal GFP-mPDE11A4 droplets form and fuse over time. Images shown capture fusion events of droplets, numbered in the order of appearance,that occured over 105 minutes. (**C**) Seven COS1 cells across five experiments confirm GFP-mPDE11A4 droplets form and fuse over time. Images shown capture fusion events that occurred over 120 minutes. (**D**) Qualitative experiments in which HT22 cells formed GFP-mPDE11A4 droplets (indicated by arrow heads) and were then imaged over time immediately prior to and following treatment with 5% 1,6-hexanediol show droplets disperse within minutes of the treatment. (**E**) Quantitative study in which HT22 cells formed GFP-mPDE11A4 droplets and then were treated with either vehicle or 5% 1,6-hexanediol confirms that GFP-mPDE11A4 droplets are reversed by 5% 1,6-hexanediol and not vehicle treatment *(n* = 4 biological replicates/group). (**F**) Quantification and representative images of HT22 cells expressing low, moderate, or high levels of GFP-mPDE11A4 protein show mPDE11A4 droplets form in a concentration-dependent manner (*n* = 4 biological replicates/group). Note, GFP alone shows no punctate clustering despite high expression throughout the cell, particularly the nucleus. (**G**) Deletion of the N-terminal intrinsically disordered region (xN-IDR) prevents PDE11A4 spherical droplets from forming (*n* = 5 biological replicates/group; representative of 3 experiments). * vs. veh/low/WT, *p* = 0.0146 to <0.0001, # vs. mod, *p* = 0.0158. oof—out of focus; mod—moderate, GFP—emerald-green fluorescent protein. Brightness/contrast/histogram stretch and/or sharpness adjusted for graphical clarity of images.

**Figure 4 cells-14-00897-f004:**
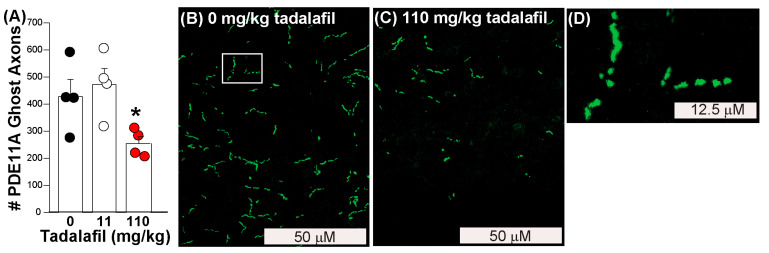
Tadalafil reverses age-related clustering of endogenous mPDE11A4 in the ventral hippocampus of old female and male mice. (**A**) Quantification of endogenous mPDE11A4 ghost axons in ventral subiculum that were selectively labeled using antibodies that recognize PDE11A4-pS1117/pS124 [9] (*n* = 2/sex/dose). Exemplar images of mPDE11A4 immunofluorescence in an (**B**) old vehicle-treated and (**C**) old tadalafil-treated mouse. (**D**) Upon higher magnification (i.e., of the area in panel A outlined by the white box), many mPDE11A4 ghost axons reveal themselves to be a linear trail of spherical dots as opposed to a continuous filament, consistent with LLPS. * Post hoc vs. 0 mg/kg tadalafil (i.e., vehicle), *p* = 0.045. Brightness, contrast and histogram stretch of images adjusted for graphical clarity.

**Table 1 cells-14-00897-t001:** PDE11A inhibitors from distinct scaffolds that were identified in screens for human PDE11A4 also inhibit mouse PDE11A4 (m11A4) cAMP- and cGMP-catalytic activity (expressed as the mean CPMs/μg protein ± SEM) in mouse HT22 hippocampal cells without changing total PDE11A4 protein levels (fold change vehicle PDE11A4 r.o.d./Ponceau S stain r.o.d.).

	GFP + 0 µM	m11A4 + 0 µM	m11A4 + 0.1 µM	m11A4 + 1 µM	m11A4 + 10 µM	m11A4 + 100 µM
**cAMP-PDE**						
tadalafil *	**117.5 ± 23.9**	3822.5 ± 97.3	3770.7 ± 460.0	3922.5 ± 182.0	**2538.8 ± 137.8**	**574.5 ± 45.0**
BC11-38	**155.0 ± 12.8**	2720.9 ± 188.2	2961.0 ± 80.3	2908.8 ± 84.9	3129.6 ± 65.6	**1882.5 ± 217.4**
**cGMP-PDE**						
tadalafil *	**52.2 ± 15.5**	3821.1 ± 175.5	3516.7 ± 441.8	3493.6 ± 239.3	2336.8 ± 239.4	**536.6 ± 50.8**
BC11-38	**54.8 ± 11.7**	2665.8 ± 130.1	2855.5 ± 259.2	2956.7 ± 126.0	3017.7 ± 61.2	**1894.3 ± 166.5**
**PDE11A4 protein**						
tadalafil *	N/A	1.00 ± 0.00	1.32 ± 0.28	1.21 ± 0.29	1.12 ± 0.17	0.94 ± 0.03
BC11-38	N/A	1.00 ± 0.05	2.39 ± 0.94	1.41 ± 0.28	1.61 ± 0.38	2.45 ± 0.55

* Tadalafil data previously reported in [27] (tadalafil called “1” therein), and shown here to illustrate the reduced potency of BC11-38 relative to tadalafil. Bolded numbers are significantly different from WT + 0 μM (i.e., vehicle-treatment), *p* < 0.05–0.0001. GFP—emerald-green fluorescent protein; m11A4—GFP-mPDE11A4.

## Data Availability

Source data are published with the paper as Appendix A.

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
