# Peer review of "Age-Related Increases in PDE11A4 Protein Expression Trigger Liquid–Liquid Phase Separation (LLPS) of the Enzyme That Can Be Reversed by PDE11A4 Small Molecule Inhibitors"

_cells, 2025, doi:10.3390/cells14120897_

Round 1
Reviewer 1 Report
Comments and Suggestions for Authors
In this work, Amurrio et al. aim to determine whether PDE11A4 clustering leads to liquid-liquid phase separation and whether PDE11A inhibitors can prevent this. They have employed a relevant overexpression system, a hippocampal cell line (HT-22 cells), to demonstrate the 11A4-LLPS and the effects of Tad and BC11-38 in reversing LLPS. Finally, they conducted an in vivo study in WT mice to show that Tad can prevent the increase in ghost axons in aging mice. Although this study is interesting and meticulously conducted, I would like to highlight a few points to improve the manuscript.
- The authors showed that 11A4-LLPS occurs without cell membranes. But this does not exclude the possibility of other proteins or scaffolds necessary for LLPS formation. Can the authors do a native gel/mass spec to show this homotypic association, or show that increasing concentrations of purified 11A4 can make this LLPS?
- Measuring PDE activity after tad and BC11-38 is interesting. Can the authors show that inhibition of PDE activity is the cause of LLPS disintegration? Can they mutate the binding domains of the inhibitors on 11A4 and still show the effect?
- The authors could overexpress cAMP or cGMP sensors and measure the cyclic nucleotides in the vicinity of LLPS and compare them with the region of the same cell without LLPS. They can do this with and without inhibitors. This can potentially explain the physiological significance of LLPS formation.
- I didn’t understand 3N and 3O. If the authors are sure that LLPS are devoid of any membrane, what is the reason behind this cyt: membrane ratio of 11A4?
- In Table 1, I see less 11A4 protein amounts, although not significant, after 10 and 100 µM of tad treatment. But the authors reported that the protein expression is unaffected by the treatment.
- Finally, the authors claim that tad treatment reduced ghost axons in the mouse hippocampus. To show that it is an 11A4-associated effect, they should do the same experiment in an 11A4-knockout system or at least show that other PDE inhibitors that affect cAMP/cGMP levels do not have the same effect as Tad.
- As the authors mentioned in the manuscript, Tad affects many systems. They should take the effort to show that Tad treatment did not significantly alter other physiological parameters.
Author Response
We thank the reviewers for their meaningful and constructive feedback! They were a joy to work with!
- Comment: The authors showed that 11A4-LLPS occurs without cell membranes. But this does not exclude the possibility of other proteins or scaffolds necessary for LLPS formation. Can the authors do a native gel/mass spec to show this homotypic association, or show that increasing concentrations of purified 11A4 can make this LLPS?
Response: We completely agree with the reviewer that the lack of membranes does not exclude the possibility that other proteins/scaffolds are necessary for LLPS formation and apologize if our wording implied otherwise. The lack of membranes simply means that the PDE11A4 droplets meet this key criterion for LLPS (that is, that droplets are membrane-less). Indeed, we report analyses from PhasePred that predict PDE11A4 undergoes both homotypic and heterotypic condensation in a context-dependent manner (see Results, starting lines 374-378). Further, we review experimental findings that support the idea that PDE11A4 may undergo both homotypic and heterotypic condensation (See Discussion, lines 673-695). We very much appreciate the reviewer’s excellent point about native gels being able to support the PhasePred predictions, and have added the following sentence to this part of the Discussion (lines 677-681), “Native gels of old mouse hippocampus and PDE11A4-transfected cell cultures exhibiting PDE11A4 droplets reveal PDE11A4 is present both as an isolated homodimer, suggesting homotypic LLPS is possible, and as part of larger macromolecular complexes, suggesting heterotypic LLPS is possible (see Figure S1B in 1).” Thanks for the excellent suggestion!
- Comment: Measuring PDE activity after tad and BC11-38 is interesting. Can the authors show that inhibition of PDE activity is the cause of LLPS disintegration? Can they mutate the binding domains of the inhibitors on 11A4 and still show the effect?
Response: We thank the reviewer for their intuitive question and clever suggestion regarding the mutation approach (wish we had thought of that earlier!). We have recently collected data that argues the PDE11A4 LLPS-dispersing effects of PDE11Ai’s are not related to PDE11A4 catalytic inhibition. To address this concern, we have added data to the paper showing that application of cAMP or cGMP analogues does not reduce PDE11A4 LLPS, and adding cAMP or cGMP blockers does not block the ability of PDE11A4i’s to reduce PDE11A4 LLPS. We have added details about the compounds to the Methods (lines 101-105) and the data to Figure 3P-S and the Results (lines 553-576). We have also revised the Discussion around potential mechanisms underlying PDE11Ai dispersal of PDE11A4 LLPS as follows (lines 793-796): “It is also not due to elevations in cAMP/cGMP levels that occur as a consequence of inhibiting PDE11A4 catalytic activity, since cAMP/cGMP analogues fail to mimic the effect of PDE11Ai’s and cAMP/cGMP antagonists fail to block the ability of PDE11Ai’s to remix PDE11A4 (Figure 3P-S).”
- Comment: The authors could overexpress cAMP or cGMP sensors and measure the cyclic nucleotides in the vicinity of LLPS and compare them with the region of the same cell without LLPS. They can do this with and without inhibitors. This can potentially explain the physiological significance of LLPS formation.
Response: We wholeheartedly agree with the reviewer that expressing cAMP/cGMP sensors in our system is a worthwhile experimental avenue to pursue in the context of explaining the physiological significance of PDE11A4 LLPS. This approach is actually part of a grant we are currently working on submitting, but at this time we would argue this work falls beyond the scope of the present manuscript and would result in significant delays in publishing our key findings here since we have never executed this technique. To address this comment, we have revised the following passage in the Discussion (lines 811-8115):
“Depending on the molecule, LLPS can sequester unneeded protein, buffer proteins (i.e., temporarily store and then release upon demand), or accelerate biochemical reactions by virtue of concentrating enzymes with substrates in membraneless organelles 2. It will be of paramount importance to future studies to parse this out as it relates to PDE11A4 LLPS specifically. For example, expressing cAMP/cGMP sensors in our model systems, particularly sensors targeted to cytosolic versus membrane compartments, may help us dissect the physiological significance of PDE11A4 LLPS.”
4. Comment: I didn’t understand 3N and 3O. If the authors are sure that LLPS are devoid of any membrane, what is the reason behind this cyt: membrane ratio of 11A4?
Response: We thank the reviewer for pointing out our lack of clarity in explaining the relevance of the cyt:membrane ratio data in the context of LLPS. In hopes of clarifying, we revised the Results as follows (lines 577-596):
“Since PDE11Ai’s reversed aging-like GFP-mPDE11A4 LLPS, we next determined if PDE11Ai’s would also reverse aging-like changes in the distribution of GFP-mPDE11A4 between the cytosolic versus membrane fractions as a second measure of a change in subcellular localization. Even though LLPS condensates themselves are membraneless, heterotypic LLPS often takes place at the plasma membrane in response to scaffolding proteins being recruited there following receptor activation3, 4. Consistent with this model, PDE11A4 is not directly inserted into the membrane but rather is indirectly associated with the membrane1 and PhasePred results above predict PDE11A4 undergoes both homotypic and heterotypic LLPS. … We see here that a 1-hour treatment of HT22 cells with 100 μM of either tadalafil (n=4 biological replicates) or BC11-38 (n=4) has an “anti-aging” effect by increasing the GFP-mPDE11A4 cytosolic:membrane ratio relative to vehicle (n=3; F(2,8)=4.99, P=0.039; Post hoc vs. vehicle: tadalafil P=0.0391, BC11-38 P=0.0389; Figure 3N-O). Thus, both the LLPS and biochemical fractionation experiments show PDE11A4 small molecule inhibitors change the subcellular localization of PDE11A4.”
5. Comment: In Table 1, I see less 11A4 protein amounts, although not significant, after 10 and 100 µM of tad treatment. But the authors reported that the protein expression is unaffected by the treatment.
Response: We apologize, but we are puzzled by the reviewer’s comment. The protein data are normalized as a fold-change of vehicle levels, making the vehicle mean 1.0. Expression of PDE11A4 with tadalafil at 10 µM is actually numerically higher than vehicle at 1.12. While 100 uM is technically lower at 0.94, this 6% decrease from vehicle levels is nowhere near statistically significant and falls far short of explaining the 85% decrease we measure in cAMP/cGMP-PDE activity. We welcome further clarification on the comment if we have failed to address the concern.
6. Comment: Finally, the authors claim that tad treatment reduced ghost axons in the mouse hippocampus. To show that it is an 11A4-associated effect, they should do the same experiment in an 11A4-knockout system or at least show that other PDE inhibitors that affect cAMP/cGMP levels do not have the same effect as Tad.
Response: We completely agree with the reviewer that the tadalafil study has its limitations and apologize for not better acknowledging them in the manuscript. Unfortunately, it is impossible to use the classic KO approach here because we are measuring PDE11A4 itself, so this classic negative control experiment is sadly not applicable in this context. The reviewer’s suggestion of testing another PDE inhibitor to show lack of effect is excellent, though. Indeed, we tested the effect of rolipram and papaverine in our cell culture model to establish specificity (Figure 3L-M), but we did not follow through in our in vivo model. Further, in this resubmission we show in vitro that the ability of the inhibitors to reverse PDE11A4 LLPS is actually independent of their catalytic effects that increase cAMP/cGMP signaling. We can let you know that we have just recently characterized 6 additional selective PDE11A inhibitors from our recently developed “MLG” and “SMQ” series 5, 6, including testing SMQ-03-20 in vivo following oral dosing, and found identical effects as those described herein 7, 8. This body of work is currently on BioRxiv and will be submitted to a peer-reviewed journal for review as soon as the present manuscript is accepted for publication.
Since we do not currently have old mice at the present time to treat with rolipram or papaverine, and fear that waiting to produce such data will lead to us being scooped by others currently pursuing this phenomena after seeing our poster at last year’s SFN conference, we have addressed this concern by adding the following paragraph to the Discussion (lines 769-790):
“Importantly, the lowest concentration of tadalafil that was effective in our HT22 cell model (i.e., 10 µM) was also effective in the brain following oral dosing. This suggests that results from our cell model can readily be used for in vivo dose selection based on brain exposures achieved following oral dosing. The fact that tadalafil reversed age-related PDE11A4 ghost axons at a 10 µM brain exposure (i.e., 110 mg/kg) but not a 1 µM brain exposure (i.e., 11 mg/kg) suggests that this effect of tadalafil is mediated via it binding to PDE11A4, as opposed to inhibiting PDE5 since much lower plasma exposures of tadalafil are required for treating erectile dysfunction9. The fact that both inhibitors reverse PDE11A4 LLPS independent of increases in cAMP or cGMP signaling also argues against tadalafil altering PDE11A4 LLPS as a consequence of inhibiting PDE5 catalytic activity. That said, we are unable to unequivocally confirm the PDE11A4 LLPS dispersing effect of tadalafil is mediated specifically by binding PDE11A4—as opposed to inhibiting PDE5 or eliciting other general physiological effects—because the endpoint is PDE11A4 itself. This leaves the classic negative control approach of testing the compound in Pde11a KO mice to establish specificity of the compound for PDE11A4 unusable. We should also note that this study is limited by the fact that we did not test rolipram or papaverine in vivo as we did in vitro to demonstrate specificity. Still, the ability of tadalafil to reverse PDE11A4 ghost axons in vivo does provide proof of principle for pursuing more selective and potent PDE11A4 inhibitors to reverse this age-related phenotype. Indeed, since such high concentrations of tadalafil may produce side effects, we are currently testing effects of orally-dosed SMQ-03-20, the first selective and potent PDE11A4 inhibitor to cross the blood-brain-barrier 5.
7. Comment: As the authors mentioned in the manuscript, Tad affects many systems. They should take the effort to show that Tad treatment did not significantly alter other physiological parameters.
Response: We thank the reviewer for this constructive feedback, and wished we had thought to measure such endpoints at the time of testing. Unfortunately, we are not in a position to repeat this experiment to collect such measurements. To address this concern, we have acknowledged our inability to separate tadalafil’s PDE11A-inhibiting from effects on other physiological endpoints with the following passage in the Discussion (lines 779-783): “That said, we are unable to unequivocally confirm the PDE11A4 LLPS dispersing effect of tadalafil is mediated specifically by binding PDE11A4—as opposed to inhibiting PDE5 or eliciting other general physiological effects—because the endpoint is PDE11A4 itself. This leaves the classic negative control approach of testing the compound in Pde11a KO mice to establish specificity of the compound for PDE11A4 unusable.”
- Pilarzyk K, Porcher L, Capell WR, Burbano SD, Davis J, Fisher JL, Gorny N, Petrolle S, Kelly MP. Conserved age-related increases in hippocampal PDE11A4 cause unexpected proteinopathies and cognitive decline of social associative memories. Aging cell. 2022;21(10):e13687. Epub 20220908. doi: 10.1111/acel.13687. PubMed PMID: 36073342; PMCID: PMC9577960.
- Alberti S, Dormann D. Liquid-Liquid Phase Separation in Disease. Annu Rev Genet. 2019;53:171-94. Epub 20190820. doi: 10.1146/annurev-genet-112618-043527. PubMed PMID: 31430179.
- Jaqaman K, Ditlev JA. Biomolecular condensates in membrane receptor signaling. Current opinion in cell biology. 2021;69:48-54. Epub 20210115. doi: 10.1016/j.ceb.2020.12.006. PubMed PMID: 33461072; PMCID: PMC8058224.
- Ditlev JA. Membrane-associated phase separation: organization and function emerge from a two-dimensional milieu. J Mol Cell Biol. 2021;13(4):319-24. doi: 10.1093/jmcb/mjab010. PubMed PMID: 33532844; PMCID: PMC8339363.
- Mahmood SU, Eberhard J, Hoffman CS, Colussi D, Gordon J, Childers W, Amurrio E, Patel J, Kelly MP, Rotella DP. First Demonstration of In Vivo PDE11A4 Target Engagement for Potential Treatment of Age-Related Memory Disorders. Journal of medicinal chemistry. 2024;67(19):17774-84. Epub 20240925. doi: 10.1021/acs.jmedchem.4c01794. PubMed PMID: 39321314.
- Mahmood SU, Lozano Gonzalez M, Tummalapalli S, Eberhard J, Ly J, Hoffman CS, Kelly MP, Gordon J, Colussi D, Childers W, Rotella DP. First Optimization of Novel, Potent, Selective PDE11A4 Inhibitors for Age-Related Cognitive Decline. Journal of medicinal chemistry. 2023;66(21):14597-608. Epub 20231020. doi: 10.1021/acs.jmedchem.3c01088. PubMed PMID: 37862143; PMCID: PMC10641827.
- Amurrio E, Patel J, Danaher M, Goodwin M, Kargbo P, Lin S, Hoffman C, Ul Mahmood S, Rotella D, Kelly M. Age-related increases in PDE11A4 protein expression trigger liquid:liquid phase separation (LLPS) of the enzyme that can be reversed by PDE11A4 small molecules inhibitors. bioRxiv. 2024:2024.10.01.616004. doi: 10.1101/2024.10.01.616004.
- Amurrio E, Patel J, Danaher M, Elzaree L, Fisher JL, Greene H, Kargbo P, Kim P, Klimentova E, Lin S, Liu Y, Abdella S, Mehboob MY, Temesgen Y, Mahmood S, Rotella DP, Kelly M. Molecular mechanisms regulating PDE11A4 age-related liquid-liquid phase separation (LLPS) and its reversal by selective, potent and orally-available PDE11A4 small molecule inhibitors both in vitro and in vivo. bioRxiv. 2025:2025.05.20.654583. doi: 10.1101/2025.05.20.654583.
- Sung HH, Lee SW. Chronic low dosing of phosphodiesterase type 5 inhibitor for erectile dysfunction. Korean journal of urology. 2012;53(6):377-85. Epub 20120619. doi: 10.4111/kju.2012.53.6.377. PubMed PMID: 22741044; PMCID: PMC3382685.
- Weeks JL, Zoraghi R, Beasley A, Sekhar KR, Francis SH, Corbin JD. High biochemical selectivity of tadalafil, sildenafil and vardenafil for human phosphodiesterase 5A1 (PDE5) over PDE11A4 suggests the absence of PDE11A4 cross-reaction in patients. International journal of impotence research. 2005;17(1):5-9. Epub 2004/11/13. doi: 10.1038/sj.ijir.3901283. PubMed PMID: 15538396.
- Ahmed NS, Gary BD, Tinsley HN, Piazza GA, Laufer S, Abadi AH. Design, synthesis and structure-activity relationship of functionalized tetrahydro-beta-carboline derivatives as novel PDE5 inhibitors. Arch Pharm (Weinheim). 2011;344(3):149-57. Epub 2011/03/09. doi: 10.1002/ardp.201000236. PubMed PMID: 21384413; PMCID: PMC4980839.
- Ceyhan O, Birsoy K, Hoffman CS. Identification of Biologically Active PDE11-Selective Inhibitors Using a Yeast-Based High-Throughput Screen. Chem Biol. 2012;19(1):155-63. Epub 2012/01/31. doi: 10.1016/j.chembiol.2011.12.010. PubMed PMID: 22284362.
Reviewer 2 Report
Comments and Suggestions for Authors
This study provides compelling evidence that age-related clustering of PDE11A4 in hippocampal "ghost axons" is driven by liquid-liquid phase separation (LLPS) and demonstrates the therapeutic potential of PDE11A inhibitors to reverse this pathology. The work is conceptually innovative, bridging biophysical mechanisms (LLPS) with age-related cognitive decline, and methodologically rigorous, combining bioinformatic predictions, in vitro cellular models, and in vivo validation. Key strengths include: (1) identification of LLPS-promoting features (e.g., N-terminal intrinsically disordered regions, prion-like domains) in PDE11A4 through sequence analysis; (2) functional validation using truncation mutants to confirm the necessity of the disordered region for LLPS; and (3) translational relevance of showing tadalafil’s efficacy in reducing ghost axons in aged mice. The conclusions are well-supported by the data, and the integration of computational and experimental approaches strengthens the mechanistic insights.
Suggested Revisions:
- Mechanistic Specificity of LLPS Drivers: While the N-terminal truncation experiment convincingly links disordered regions to LLPS, the contribution of π-π interactions or prion-like domains remains correlative. Mutational analysis targeting specific motifs (e.g., aromatic residues for π-π interactions) would strengthen causal evidence.
- Pharmacological Selectivity: Although tadalafil and BC11-38 are described as PDE11A inhibitors, their selectivity over other PDE isoforms (e.g., PDE5 for tadalafil) is not addressed. Co-treatment with isoform-specific inhibitors or PDE11A-knockout models would clarify target specificity.
- Clinical Relevance of Dosing: The high tadalafil dose (110 mg/kg) achieving 10 μM brain exposure far exceeds human clinical doses (~1.8 μM plasma Cmax for erectile dysfunction). A discussion of dose translation challenges, toxicity risks, or strategies to enhance brain delivery (e.g., nanoparticle formulations) would contextualize therapeutic feasibility.
Author Response
We thank the reviewers for their meaningful and constructive feedback! They were a joy to work with!
- Comment: Mechanistic Specificity of LLPS Drivers: While the N-terminal truncation experiment convincingly links disordered regions to LLPS, the contribution of π-π interactions or prion-like domains remains correlative. Mutational analysis targeting specific motifs (e.g., aromatic residues for π-π interactions) would strengthen causal evidence.
Response: We thank the reviewer for this excellent recommendation. Indeed, these are amongst future experiments in our grant proposal currently in prep! Given these studies will take months to complete, coupled with our fear of being scooped by others currently pursuing PDE11A4 LLPS after seeing our poster at last year’s SFN conference, we are hoping the reviewer will be satisfied for now with the following sentence we have added to the Discussion (lines 685-687): “It will be of great interest to future experiments to more selectively probe the functional relevance of the pro-LLPS sequence motifs that we identified within the PDE11A4 N-terminal IDR (Figure 1; e.g., prion-like domains and π-π interactions).”
- Comment: Pharmacological Selectivity: Although tadalafil and BC11-38 are described as PDE11A inhibitors, their selectivity over other PDE isoforms (e.g., PDE5 for tadalafil) is not addressed. Co-treatment with isoform-specific inhibitors or PDE11A-knockout models would clarify target specificity.
Response: We agree that establishing specificity is important for these studies, and we have struggled with how to best address this since the endpoint we are examining is PDE11A4 itself. Meaning, the classic drug + Pde11a KO approach is not viable here since there is no PDE11A4 protein to measure. We attempted to assess specificity by testing BC11-38, which is a highly selective PDE11A inhibitor (>100-fold selective versus PDE1-10). Also, we attempted to address specificity with the rolipram and papaverine experiments that we conducted in vitro, which showed no effect. We can let you know that we have just recently characterized 6 additional selective PDE11A inhibitors from our recently developed “MLG” and “SMQ” series 5, 6, including testing SMQ-03-20 in vivo following oral dosing, and found identical effects as those described herein 7, 8. This body of work is currently on BioRxiv and will be submitted to a peer-reviewed journal for review as soon as the present manuscript is accepted for publication.
To address this concern, we revised the Results section to provide additional background details regarding selectivity of tadalafil and BC11-38 (505-512):
“Here we report findings with molecules from 2 scaffolds that inhibit PDE11A4. The first scaffold is represented by tadalafil (Figure 3B), a PDE5A inhibitor approved for treatment of erectile dysfunction and benign prostatic hypertrophy that is well known to potently inhibit PDE11A4, albeit at higher concentrations10, 11. For example, in a pure enzyme assay, tadalafil inhibits PDE5 cGMP-PDE activity with an IC50 of 0.007 µM and PDE11A4 cGMP-PDE activity with an IC50 of 0.05 µM11. The second scaffold is represented by BC11-38 (Figure 3B), a PDE11A-selective inhibitor (0.28 µM IC50 in a pure enzyme assay), with >100-fold selectivity for PDE11A4 versus PDE1-1012.”
We also added the following paragraph to the Discussion (lines 769-790):
“Importantly, the lowest concentration of tadalafil that was effective in our HT22 cell model (i.e., 10 µM) was also effective in the brain following oral dosing. This suggests that results from our cell model can readily be used for in vivo dose selection based on brain exposures achieved following oral dosing. The fact that tadalafil reversed age-related PDE11A4 ghost axons at a 10 µM brain exposure (i.e., 110 mg/kg) but not a 1 µM brain exposure (i.e., 11 mg/kg) suggests that this effect of tadalafil is mediated via it binding to PDE11A4, as opposed to inhibiting PDE5 since much lower plasma exposures of tadalafil are required for treating erectile dysfunction9. The fact that both inhibitors reverse PDE11A4 LLPS independent of increases in cAMP or cGMP signaling also argues against tadalafil altering PDE11A4 LLPS as a consequence of inhibiting PDE5 catalytic activity. That said, we are unable to unequivocally confirm the PDE11A4 LLPS dispersing effect of tadalafil is mediated specifically by binding PDE11A4—as opposed to inhibiting PDE5 or eliciting other general physiological effects—because the endpoint is PDE11A4 itself. This leaves the classic negative control approach of testing the compound in Pde11a KO mice to establish specificity of the compound for PDE11A4 unusable. We should also note that this study is limited by the fact that we did not test rolipram or papaverine in vivo as we did in vitro to demonstrate specificity. Still, the ability of tadalafil to reverse PDE11A4 ghost axons in vivo does provide proof of principle for pursuing more selective and potent PDE11A4 inhibitors to reverse this age-related phenotype. Indeed, since such high concentrations of tadalafil may produce side effects, we are currently testing effects of orally-dosed SMQ-03-20, the first selective and potent PDE11A4 inhibitor to cross the blood-brain-barrier 5.
3. Comment: Clinical Relevance of Dosing: The high tadalafil dose (110 mg/kg) achieving 10 μM brain exposure far exceeds human clinical doses (~1.8 μM plasma Cmax for erectile dysfunction). A discussion of dose translation challenges, toxicity risks, or strategies to enhance brain delivery (e.g., nanoparticle formulations) would contextualize therapeutic feasibility.
Response: We thank the reviewer for this insightful and constructive comment. We had not intended to suggest that tadalafil itself would be used as a therapeutic in the context of PDE11A4 LLPS for all the reasons the reviewer mentioned, in particular side effect liability. Rather, we see this data as a proof of concept for exploring more selective PDE11A4 inhibitors for this purpose. To address this concern, we added the following passage to the Discussion (starting on line 748):
“Still, the ability of tadalafil to reverse PDE11A4 ghost axons in vivo does provide proof of principle for pursuing more selective and potent PDE11A4 inhibitors to reverse this age-related phenotype. Indeed, since such high concentrations of tadalafil may produce side effects, we are currently testing effects of orally-dosed SMQ-03-20, the first selective and potent PDE11A4 inhibitor to cross the blood-brain-barrier 5.”
- Pilarzyk K, Porcher L, Capell WR, Burbano SD, Davis J, Fisher JL, Gorny N, Petrolle S, Kelly MP. Conserved age-related increases in hippocampal PDE11A4 cause unexpected proteinopathies and cognitive decline of social associative memories. Aging cell. 2022;21(10):e13687. Epub 20220908. doi: 10.1111/acel.13687. PubMed PMID: 36073342; PMCID: PMC9577960.
- Alberti S, Dormann D. Liquid-Liquid Phase Separation in Disease. Annu Rev Genet. 2019;53:171-94. Epub 20190820. doi: 10.1146/annurev-genet-112618-043527. PubMed PMID: 31430179.
- Jaqaman K, Ditlev JA. Biomolecular condensates in membrane receptor signaling. Current opinion in cell biology. 2021;69:48-54. Epub 20210115. doi: 10.1016/j.ceb.2020.12.006. PubMed PMID: 33461072; PMCID: PMC8058224.
- Ditlev JA. Membrane-associated phase separation: organization and function emerge from a two-dimensional milieu. J Mol Cell Biol. 2021;13(4):319-24. doi: 10.1093/jmcb/mjab010. PubMed PMID: 33532844; PMCID: PMC8339363.
- Mahmood SU, Eberhard J, Hoffman CS, Colussi D, Gordon J, Childers W, Amurrio E, Patel J, Kelly MP, Rotella DP. First Demonstration of In Vivo PDE11A4 Target Engagement for Potential Treatment of Age-Related Memory Disorders. Journal of medicinal chemistry. 2024;67(19):17774-84. Epub 20240925. doi: 10.1021/acs.jmedchem.4c01794. PubMed PMID: 39321314.
- Mahmood SU, Lozano Gonzalez M, Tummalapalli S, Eberhard J, Ly J, Hoffman CS, Kelly MP, Gordon J, Colussi D, Childers W, Rotella DP. First Optimization of Novel, Potent, Selective PDE11A4 Inhibitors for Age-Related Cognitive Decline. Journal of medicinal chemistry. 2023;66(21):14597-608. Epub 20231020. doi: 10.1021/acs.jmedchem.3c01088. PubMed PMID: 37862143; PMCID: PMC10641827.
- Amurrio E, Patel J, Danaher M, Goodwin M, Kargbo P, Lin S, Hoffman C, Ul Mahmood S, Rotella D, Kelly M. Age-related increases in PDE11A4 protein expression trigger liquid:liquid phase separation (LLPS) of the enzyme that can be reversed by PDE11A4 small molecules inhibitors. bioRxiv. 2024:2024.10.01.616004. doi: 10.1101/2024.10.01.616004.
- Amurrio E, Patel J, Danaher M, Elzaree L, Fisher JL, Greene H, Kargbo P, Kim P, Klimentova E, Lin S, Liu Y, Abdella S, Mehboob MY, Temesgen Y, Mahmood S, Rotella DP, Kelly M. Molecular mechanisms regulating PDE11A4 age-related liquid-liquid phase separation (LLPS) and its reversal by selective, potent and orally-available PDE11A4 small molecule inhibitors both in vitro and in vivo. bioRxiv. 2025:2025.05.20.654583. doi: 10.1101/2025.05.20.654583.
- Sung HH, Lee SW. Chronic low dosing of phosphodiesterase type 5 inhibitor for erectile dysfunction. Korean journal of urology. 2012;53(6):377-85. Epub 20120619. doi: 10.4111/kju.2012.53.6.377. PubMed PMID: 22741044; PMCID: PMC3382685.
- Weeks JL, Zoraghi R, Beasley A, Sekhar KR, Francis SH, Corbin JD. High biochemical selectivity of tadalafil, sildenafil and vardenafil for human phosphodiesterase 5A1 (PDE5) over PDE11A4 suggests the absence of PDE11A4 cross-reaction in patients. International journal of impotence research. 2005;17(1):5-9. Epub 2004/11/13. doi: 10.1038/sj.ijir.3901283. PubMed PMID: 15538396.
- Ahmed NS, Gary BD, Tinsley HN, Piazza GA, Laufer S, Abadi AH. Design, synthesis and structure-activity relationship of functionalized tetrahydro-beta-carboline derivatives as novel PDE5 inhibitors. Arch Pharm (Weinheim). 2011;344(3):149-57. Epub 2011/03/09. doi: 10.1002/ardp.201000236. PubMed PMID: 21384413; PMCID: PMC4980839.
- Ceyhan O, Birsoy K, Hoffman CS. Identification of Biologically Active PDE11-Selective Inhibitors Using a Yeast-Based High-Throughput Screen. Chem Biol. 2012;19(1):155-63. Epub 2012/01/31. doi: 10.1016/j.chembiol.2011.12.010. PubMed PMID: 22284362.
Round 2
Reviewer 1 Report
Comments and Suggestions for Authors
The authors have answered my comments enough.
Reviewer 2 Report
Comments and Suggestions for Authors
This revised manuscript can be accepted for publication.